# Computational conjugate adaptive optics microscopy for longitudinal through-skull imaging of cortical myelin

Yongwoo Kwon[1,2,5], Jin Hee Hong [1,2,5], Sungsam Kang [1,2,5], Hojun Lee[1,2], Yonghyeon Jo [1,2], Ki Hean Kim [3], Seokchan Yoon [1,2,4] ✉ & Wonshik Choi [1,2] ✉

Myelination processes are closely related to higher brain functions such as learning and memory. While their longitudinal observation has been crucial to understanding myelin-related physiology and various brain disorders, skull opening or thinning has been required to secure clear optical access. Here we present a high-speed reflection matrix microscope using a light source with a wavelength of 1.3 μm to reduce tissue scattering and aberration. Furthermore, we develop a computational conjugate adaptive optics algorithm designed for the recorded reflection matrix to optimally compensate for the skull aberrations. These developments allow us to realize label-free longitudinal imaging of cortical myelin through an intact mouse skull. The myelination processes of the same mice were observed from 3 to 10 postnatal weeks to the depth of cortical layer 4 with a spatial resolution of 0.79 μm. Our system will expedite the investigations on the role of myelination in learning, memory, and brain disorders.

Myelin is a complex cellular structure in which a lipid-rich multi-layered membrane tightly wraps around the axons to provide electrical insulation and metabolic support. Myelin enables salta-tory axonal conduction that substantially increases the neural processing speed and energy efficiency[1]. Myelin is mainly concentrated in the white matter in the central nervous system and widespread in the gray matter. In particular, the myelin patterns are dynamic in the mouse cortex with aging and learning[2,3]. It is also proposed that the myelination of pyramidal neurons located deep in the cortical layer can be an essential substrate for neuroplasticity[4]. These results suggest that myelin could play a crucial role in higher brain functions, such as learning and memory. To further investigate myelin-related physiology and brain dis-order, it is critical to observe the myelination processes from the early developmental stages and for an extended period with mini-mal impairments to the subjects.

The longitudinal imaging of cortical myelin in live animals requires high resolution and deep-tissue imaging capability because the cortical myelin is very thin, ~1 μm in diameter, and it is distributed from the shallow to deep layers of the brain cortex. Fluorescence imaging modalities relying on either external dye injection or genetic labeling have been used to visualize myeli-nating cells[5]. However, exogenous dyes cannot penetrate to a sufficient depth, and sparse labeling is usually implemented in genetic labeling to minimize background fluorescence in deep-tissue imaging, which leads to the visualization of only a small fraction of mature myelination. Label-free imaging modalities, such as optical coherence microscopy (OCM)[6,7] combining the time-gating and confocal gating, spectral confocal reflectance microscopy[8], third-harmonic generation microscopy[9], and coher-ent Raman imaging[10] have drawn special attention as they are free from these issues. However, all these methods require either the

[1]Center for Molecular Spectroscopy and Dynamics, Institute for Basic Science, Seoul 02841, Korea. [2]Department of Physics, Korea University, Seoul 02855, Korea. [3]Department of Mechanical Engineering, Pohang University of Science and Technology, Pohang 37673, Korea. [4]School of Biomedical Convergence Engineering, Pusan National University, Yangsan 50612, Korea. [5]These authors contributed equally: Yongwoo Kwon, Jin Hee Hong, Sungsam Kang. ✉e-mail: sc.yoon@pusan.ac.kr; wonshik@korea.ac.kr

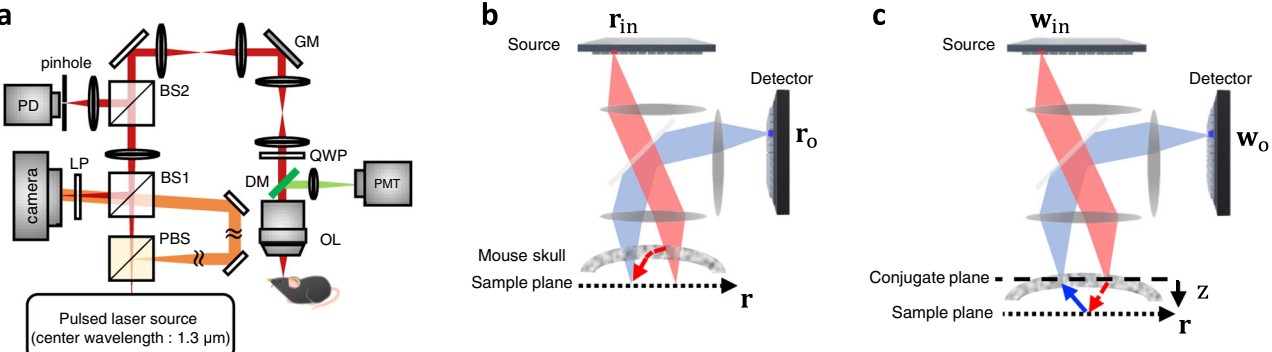

**Fig. 1 | Description of the imaging system, and the illustration of the basis conversion. a** Layout of the 1.3 μm reflection matrix microscope system. BS beam splitter (50:50), PBS polarizing beam splitter, PD photodetector, GM galvanometer mirrors for scanning a focused illumination along *x* and *y* directions at the sample, LP linear polarizer, QWP quarter-wave plate, DM dichroic mirror, OL objective lens, and PMT photomultiplier tube. **b** Schematic description of the space-domain reflection matrix $\boldsymbol{R}$. The focused beam is illuminated from $\mathbf{r}_{in}$ conjugate to the sample plane, and the beam scattered back from the sample is detected at $\mathbf{r}_o$ conjugate to the sample plane. **c** Schematic description of a conjugate plane reflection matrix $\boldsymbol{R}_{con}$. The focused beam is illuminated from $\mathbf{w}_{in}$ conjugate to the skull, and the wave scattered via the sample plane is detected at $\mathbf{w}_o$ conjugate to the skull. Note that this schematic is a hypothetical experimental setup describing the basis of $\boldsymbol{R}_{con}$, not the real experimental setup.

removal of the skull for the implantation of a cranial window or thinning of the skull to secure clear optical access. These procedures can cause inadvertent mechanical trauma[11]. Although the thinned skull with glass reinforcement can significantly minimize inflammation[12], the outcome of skull opening and thinning depends highly on the surgical skill. They can particularly be disruptive in young subjects, thereby hampering the investigation of the early developmental stages. Recently, we reported a label-free reflectance imaging method with a source wavelength of 0.9 μm[13] that is capable of imaging the myelin in cortex layer 1 of the mouse brain by correcting cranial aberrations. Due to the strong scattering and complex aberration by the mouse skull, the imaging fidelity was not high enough to reach cortical layer 4 and allow a longitudinal study.

Here, we report the longitudinal imaging of cortical myelin through an intact mouse skull to the depth of cortical layer 4 (~650 μm below the dura matter). We constructed a high-speed reflection matrix microscope (RMM) using a light source with a wavelength of 1.3 μm, instead of 0.9 μm, to reduce tissue scattering and aberration[14,15]. Furthermore, we developed a computational conjugate adaptive optics (AO) algorithm termed conjugate closed-loop accumulation of single scattering (conjugate-CLASS) to optimally compensate for the skull aberrations. In comparison with the previously developed pupil-CLASS[13,16], the conjugate-CLASS substantially enlarges the isoplanatic patch size, thereby enhancing the image reconstruction fidelity with reduced computational cost. Compared with the hardware conjugate AOs[17–19], this algorithm is flexible in the choice of the conjugate plane in the post-processing steps and can remove much more complex aberrations. These combined hardware and software improvements substantially raised the fidelity of the through-skull imaging, making it possible to visualize cortical layer 4 with high spatial resolution. We conducted label-free reflectance imaging of the individual myelin segments in the same area in the somatosensory cortex of the same mouse from its weaning stages (postnatal days 23–28 (P23–P28)) to the stages close to adulthood (P56–P70). In doing so, we quantified the growth rate of the myelination processes in cortical layer 1 and observed the emergence of new myelin segments in cortical layer 4 with development.

## Results

### 1.3-μm reflection matrix microscope setup and data acquisition procedure

The backbone of the 1.3-μm RMM system is an interferometric confocal microscope, but we replaced the confocal pinhole and

photodetector with a camera to measure the phase and amplitude of all the elastic backscattering signals from the sample (Fig. 1a) (see Supplementary Fig. 1 for detailed experimental setup). A wavelength-tunable pulsed laser (INSIGHT X3, Spectra physics) with a bandwidth of 19 nm at the wavelength of 1.3 μm was used as the light source to provide a coherence gating window (or time-gating window) of 25 μm. The output beam from the laser was delivered to the sample via two galvanometer scanning mirrors for raster scanning with a focused illumination at the sample plane. A long-working-distance objective lens (XLPLN25XWMP2, Olympus, ×25, 1.05 NA) was used to ensure a high spatial resolving power for obtaining images from areas deep within the tissues. The backscattering signal from the sample was captured by the same objective lens and then delivered to the camera after being descanned by the same scanning mirrors. A fast InGaAs camera (Cheetah 800, Xenics, 6.8 kHz frame rate) was placed at a plane conjugate to the sample to record the interference image formed by the elastic backscattering signal from the sample and the reference beams. The off-axis interferogram was processed to obtain the phase and amplitude maps of the sample waves for each illumination position. The back-reflection noise from various optical components in the present setup was significantly large owing to the relatively low quality of the anti-reflection coating. A combination of a linear polarizer and a quarter-wave plate, centered at a wavelength of 1.31 μm, was installed to minimize the stray reflections from the optics. Our system can also be used to record confocal fluorescence and second harmonic generation (SHG) images by installing a dichroic mirror and a photomultiplier tube (H5784-20, Hamamatsu Photonics). The sample stage, scanning mirrors, data acquisition board, and camera were controlled via MATLAB.

We validated the performance of the conjugate-CLASS algorithm for a resolution target under an excised mouse skull. A time-gated reflection matrix $\boldsymbol{R}$ was recorded with respect to the sample plane set by the objective focus, whose lateral coordinates are $\mathbf{r} = (x, y)$ (Fig. 1b). A focused illumination is sent to $\mathbf{r}_{in} = (x_{in}, y_{in})$ conjugate to the sample plane, and the complex-field map of the backscattered wave $E(\mathbf{r}_o; \mathbf{r}_{in})$ is measured at detector position $\mathbf{r}_o = (x_o, y_o)$ conjugate to the sample plane. Figure 2a shows $E(\mathbf{r}_o - \mathbf{r}_{in}; \mathbf{r}_{in})$ for a few representative $\mathbf{r}_{in}$. The center of each image corresponds to $\mathbf{r}_{in}$ due to the descanning by the scanning mirror in the reflection beam path. The $E(\mathbf{r}_o; \mathbf{r}_{in})$ constitutes the matrix element of $\boldsymbol{R}$ at the column and row indices corresponding to $\mathbf{r}_{in}$ and $\mathbf{r}_o$,

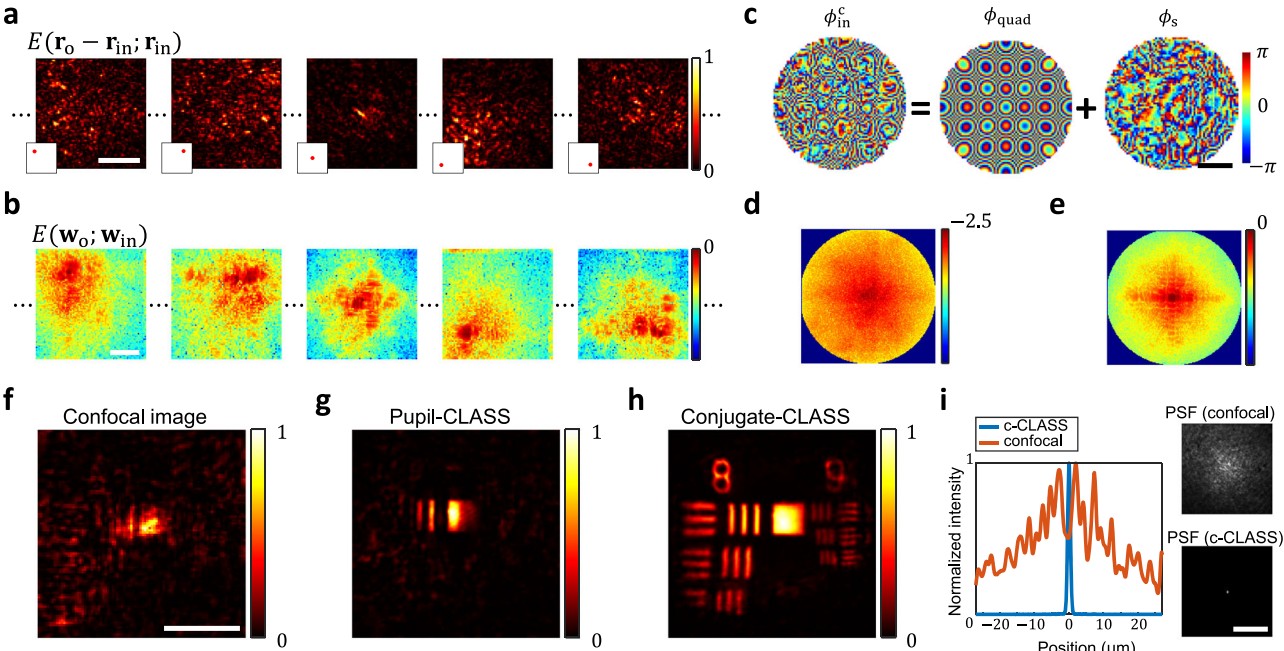

**Fig. 2 | The working principle of the conjugate-CLASS algorithm. a** A set of recorded electric field images $E(\mathbf{r}_o - \mathbf{r}_{in}; \mathbf{r}_{in})$ of the reflected wave in the camera plane for various $\mathbf{r}_{in}$ (only the amplitude maps are shown). These images constitute the reflection matrix $\mathbf{R}$. A USAF resolution target was placed under the skull. The inset images indicate the illumination points $\mathbf{r}_{in}$ within the field of illumination. Color bar: normalized amplitude. **b** Each image in **a** was converted into $E(\mathbf{w}_o; \mathbf{w}_{in})$ after the basis conversion (only the amplitude maps are shown). These images constitute the $\mathbf{R}_{con}$. Color bar: log-scale of the normalized amplitude. **c** Aberration correction map $\phi_{in}^c$ obtained by conjugate-CLASS algorithm. It is decomposed into the quadratic phase $\phi_{quad} = (k/2z)|\mathbf{w}_{in}|^2$ and the skull-induced aberration $\phi_s$. Color bar: phase in radians. **d** $\widetilde{O}_M$ before conjugate-CLASS correction. **e** $\widetilde{O}_M$ after conjugate-CLASS correction. Color bars in **d**, **e**: log-scale of the normalized intensity with respect to the maximum intensity in **e**. **f** Time-gated confocal image. **g** Aberration-corrected image by the pupil-CLASS method. **h** conjugate-CLASS image obtained by the inverse Fourier transform of $\widetilde{O}_M$ in **e**. Color bars in **f**–**h**: normalized intensity. **i** PSFs before and after aberration correction obtained from the reflection matrix. They correspond to the convolution of input and output PSFs, thereby accounting for the roundtrip aberrations. Line plots along the center pixel line in the PSF images are shown on the lefthand. c-CLASS conjugate-CLASS. Scale bar: 20 μm (**a**, **f**, **i**), 100 μm (**b**, **c**).

respectively. The field of detection at the camera for each focused illumination was typically $25 \times 25\ \mu m^2$, and it took 2.6 s to scan the focus across a field of illumination of $80 \times 80\ \mu m^2$ with a scanning step size of $0.65\ \mu m$ (see Supplementary section 1.2 for detailed data acquisition time). The decorrelation time due to the motion of the living mice was much longer than the recording time of each reflection matrix (Supplementary section 1.3).

## Conjugate-CLASS algorithm

In our previous study, we applied an algorithm, called a CLASS, to $\mathbf{R}$ for dealing with the aberrations in the pupil plane[20]. This is well-suited for samples whose aberration sources are far from the sample plane. In the case of through-skull imaging, the mouse skull is in proximity to the sample plane, causing a position-dependent aberration on the basis of pupil correction. Therefore, the aberrations must be addressed for each small subregion, which undermines the fidelity of the aberration correction. In conjugate-CLASS, we consider that the aberration layer is located at a plane conjugate to the skull and computationally change the basis of the reflection matrix from the sample plane to the plane where the skull is located. This made it possible to compensate for the skull-induced aberrations over a large field of view with enhanced image reconstruction fidelity and quality.

For applying the conjugate-CLASS algorithm, we converted $E(\mathbf{r}_o; \mathbf{r}_{in})$ into $E(\mathbf{w}_o; \mathbf{w}_{in})$ constituting a conjugate plane reflection matrix $\mathbf{R}_{con}$ by the basis conversion from $(\mathbf{r}_o, \mathbf{r}_{in})$ to $(\mathbf{w}_o, \mathbf{w}_{in})$. Here $\mathbf{w}_{in} = (u_{in}, v_{in})$ and $\mathbf{w}_o = (u_o, v_o)$ are respectively the illumination and detection positions conjugate to the skull located at a distance $z$ from the sample plane (Fig. 1c). The basis conversion is realized by applying free-space propagation (see Supplementary section 2.1 for

details). Figure 2b shows $E(\mathbf{w}_o; \mathbf{w}_{in})$ for a few representative $\mathbf{w}_{in}$. Under Fresnel approximation, $E(\mathbf{w}_o; \mathbf{w}_{in})$ is written as

$$E(\mathbf{w}_o; \mathbf{w}_{in}) = -\left(e^{2ikz}/\lambda^2 z^2\right)e^{i\phi_{in}(\mathbf{w}_{in})}\widetilde{O}_M\left(k(\mathbf{w}_o + \mathbf{w}_{in})/z\right)$$
$$e^{i\phi_o(\mathbf{w}_o)} + E_M(\mathbf{w}_o; \mathbf{w}_{in}) \qquad (1)$$

Here, $\phi_{in}(\mathbf{w}_{in}) = \phi_s(\mathbf{w}_{in}) + \frac{k}{2z}|\mathbf{w}_{in}|^2$ and $\phi_o(\mathbf{w}_o) = \phi_s(\mathbf{w}_o) + \frac{k}{2z}|\mathbf{w}_o|^2$. $\phi_s$ is the aberration map of the skull at the conjugate plane ($k = 2\pi/\lambda$ with $\lambda$ the wavelength of the light source). In theory, $\phi_{in}$ and $\phi_o$ are the same due to the reciprocity, but we treat them independently to find $\phi_{in}$ and $\phi_o$ iteratively by exploiting the wave correlation in the reflection matrix. This also accounts for the potential mismatch between $\phi_{in}$ and $\phi_o$ due to experimental imperfections such as a slight mismatch between the illumination and detection pathways. $\widetilde{O}_M$ is the 2D Fourier transform of $O_M(\mathbf{r}) = O(\mathbf{r})\exp\{i(k/z)|\mathbf{r}|^2\}$, where $O(\mathbf{r})$ is the amplitude reflectance of the target object at the sample plane (see Supplementary section 2.2 for the detailed derivation of Eq. (1)). $E_M(\mathbf{w}_o; \mathbf{w}_{in})$ is the multiple scattering noise from the skull and brain tissues. We developed an algorithm that finds $\phi_{in}(\mathbf{w}_{in})$, $\phi_o(\mathbf{w}_o)$, and $\widetilde{O}_M$ from $\mathbf{R}_{con}$ by exploiting the shift-invariance of $\widetilde{O}_M\left(k(\mathbf{w}_o + \mathbf{w}_{in})/z\right)$ in $E(\mathbf{w}_o; \mathbf{w}_{in})$ with respect to $\mathbf{w}_{in}$ (see Methods and Supplementary section 2.3 for details). Figure 2c shows the obtained $\phi_{in}(\mathbf{w}_{in})$, which is decomposed into the quadratic phase $\phi_{quad} = (k/2z)|\mathbf{w}_{in}|^2$ and the skull-induced aberration $\phi_s$. $\widetilde{O}_M$ obtained before and after the conjugate-CLASS correction are shown in Figs. 2d and 2e, respectively. Since the spectrum is accumulated in phase after the aberration correction, the signal intensity of $\widetilde{O}_M$ is enhanced almost 280-fold compared to that before correction. Figure 2h shows the conjugate-CLASS image reconstructed by the

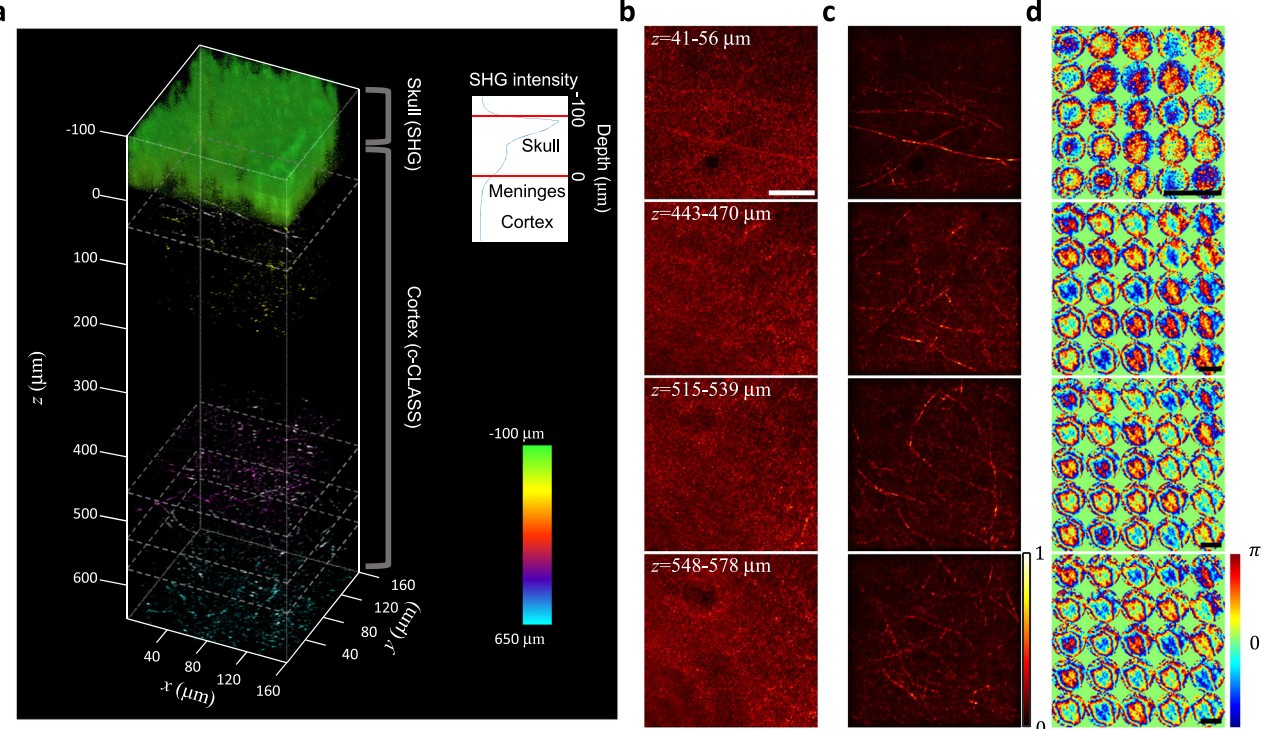

**Fig. 3 | In vivo reflectance images of a mouse brain down to cortical layer 4 through an intact skull. a** 3D-rendered through-skull image. The SHG images were used to visualize the mouse skull (colored in green). The depth-dependent intensity was plotted to estimate the skull thickness (inset). The depth of the dura matter was set as $z = 0\,\mu m$. Cortical layer 1 (up to -100 μm) and cortical layer 4 (400–650 μm) were mainly imaged since the myelinated axons are distributed laterally at these layers. Color bar: depth. **b** Conventional OCM images at representative depths displayed with gray-dotted boxes in **a**. Scale bar: 40 μm. **c** Aberration-corrected images at the same depths as those shown in **b**. Individual images are MIPs in the depth ranges of 41–56, 443–470, 515–539, and 548–578 μm. Each image was normalized by its maximum intensity. Color bar: normalized intensity. **d** Aberration maps of the subregions in **c**. Color bar: phase in radians. The aberration maps represent the spatial phase retardation at the plane conjugate to the skull. Only the output aberration maps are shown here, and the associated input aberrations are shown in Supplementary Fig. 4. Scale bars: 800 μm. Note that the diameter of the aberration map increases with depth due to illumination/detection geometry (Supplementary section 1.6).

inverse Fourier transform of the aberration-corrected $\widetilde{O}_M$ in Fig. 2e, where a clean image of the USAF target is visible over the entire field of view. On the contrary, the time-gated confocal image equivalent to the OCM image (Fig. 2f) obtained from the diagonal elements of $\boldsymbol{R}$ didn't show any structure due to the pronounced aberrations. The conjugate-CLASS algorithm converted the confocal image's substantially broadened point spread function (PSF) (~30 μm in full width at half maximum (FWHM)) into a sharp PSF (0.66 μm in FWHM) (Fig. 2i).

We compared the performance of the conjugate-CLASS with pupil-CLASS. The pupil-CLASS image shown in Fig. 2g visualizes the target structures to a certain extent, but the image quality was improved only in the small central region. This is mainly due to position-dependent aberration on the basis of pupil correction. We found that the isoplanatic patch size, where the aberration stays similar, is $15 \times 15\,\mu m^2$ in the through-skull imaging (Supplementary Fig. 8). On the contrary, the conjugate-CLASS image compensates for the skull-induced aberrations over the field of view as large as $80 \times 80\,\mu m^2$ with an enhanced image reconstruction fidelity and quality (Fig. 2h, and Supplementary Figs. 8 and 9). It should be noted that the conjugate AO has been employed in the hardware-based AO microscopy[17–19], wherein the wavefront shaping device is placed at a plane conjugate to the skull. Using the reflection matrix, we could realize the same concept in computational post-processing. In hardware conjugate AO, the wavefront shaping device should carefully match its conjugate plane to the skull during the experiment. Furthermore, time-consuming feedback optimization should be performed prior to the image acquisition. On the contrary, our conjugate-

CLASS can freely choose the optimal conjugate plane after the data acquisition and administrate a large number of iterations computationally to cope with much more complex aberrations.

## In vivo through-skull imaging
We performed the through-skull deep-brain imaging up to cortical layer 4 for a 5-week-old mouse (Fig. 3, Supplementary Movie 1). The SHG images were acquired to determine the skull thickness (~100 μm) (Fig. 3a). The depth of dura matter was set to 0 μm, and the SHG signal was displayed in the depth between −100 μm to 0 μm. Then, the time-gated reflection matrices were recorded over a field of view of $160 \times 160\,\mu m^2$ and in the depth range from 0 μm to 650 μm beneath the dura at a depth interval of 3 μm. This depth range covers the cortical layers 1–4. In each recording, the reference beam path was adjusted based on the average refractive index of the mouse brain tissue. The full recorded field of view was divided into $5 \times 5$ subregions, and the conjugate-CLASS algorithm was applied to each submatrix to identify the aberrations at the corresponding subregion (Fig. 3d). To minimize the image contrast mismatch at the edges of the subregions, the area of individual subregions was chosen to be $64 \times 64\,\mu m^2$ (Supplementary section 1.4), and then the reconstructed images were overlapped with each other in the ratio of 50%. The subregion area of $64 \times 64\,\mu m^2$ processed by the conjugate-CLASS algorithm was much larger than the isoplanatic patch size ($15 \times 15\,\mu m^2$) used in the previous pupil-plane CLASS (Supplementary Fig. 8). It should be noted that the aberration maps indicate the phase delay induced by the skull at the conjugate plane. Considering the illumination/detection geometry,

the area of the aberration map becomes larger as the imaging depth increases (see the scale bars in Fig. 3d) (Supplementary section 1.6). The aberration-corrected images from all the subregions were merged to form a full-field image at each depth, and a 3D-rendered volumetric image was reconstructed from all the depth-dependent images (Fig. 3a). For a few representative depths, the aberration-corrected images were displayed using a maximum intensity projection (MIP) over a depth range of ±15 µm (Fig. 3c, Supplementary Movies 2 and 3). As a point of reference, the conventional OCM images obtained from the same reflection matrices are shown in Fig. 3b.

Due to severe skull-induced aberrations, the conventional OCM displayed the microstructures at cortical layer 1 (0–100 µm) only vaguely and completely lost its resolving power as the depth was further increased. After compensating for the aberrations, numerous myelin fibers up to cortical layer 4 were clearly resolved (Fig. 3c). We proved that the fibril structures in the reflectance image were myelinated axons, via a separate ex vivo fluoromyelin-stained fluorescence imaging (Supplementary Fig. 12). The thinnest myelin observed at a depth of 518 µm beneath the dura (cortical layer 4) was 0.79 µm in diameter (Supplementary Fig. 13). We observed that the signal enhancement by the aberration correction estimated by the inverse of the Strehl ratio grew steadily with depth (Supplementary Figs. 11c, d). The signal enhancement at cortical layer 4 was 300–400 times. The PSF width was broadened by 88 times (~28.6 µm in FWHM) relative to the diffraction limit at cortical layer 4, supporting the difficulty in reaching cortical layer 4 via the intact skull.

### Longitudinal imaging for monitoring the development of myelination

We conducted a longitudinal observation of a mouse brain up to cortical layer 4 through the intact skull, from its weaning period to the early adult stage (Fig. 4). We could start the investigation at an early stage, as young as three weeks after the birth, because the intact skull imaging is a non-invasive technique that does negligible harm to the subject. The same region of the mouse brain was repeatedly imaged by keeping track of the blood vessel structures beneath the skull. In each imaging session, a volumetric image was acquired from the surface of the skull up to a depth of 650 µm below the dura. However, the analysis was mainly focused on cortical layers 1 and 4, where the axons stretch laterally. Figure 4a shows the MIP images of cortical layer 1 recorded at P26, P33, and P47 over a field of view of 400 × 400 µm² and in a depth range between ~40 µm to ~90 µm below the dura. Notably, different depth ranges were chosen for comparing the myelin at different ages because of the growth of the mouse skull and brain. A drastic increase in the density of myelin was observed, especially at a very early stage of the lifetime of the mice. The high-resolution images allowed us to trace the growth of the individual myelination processes. For example, the yellow and white arrowheads in Fig. 4a indicate the newly emerged myelin segments that did not appear in the preceding stages.

A similar progression of the myelination processes was consistently observed in another mouse (Fig. 4b, c). In this case, the growth of myelination was monitored at P33, P40, P47, and P74. In Fig. 4b, the myelin segments indicated by the white and yellow arrowheads at P74 are absent at P33. Figure 4c shows the gradual elongation of these myelin segments. We measured the lengths of individual myelin segments derived from the MIP images shown in Fig. 4b (Fig. 4d). The median values of myelin segment length increased with age. In addition, we measured the sum of all the myelin segment lengths per unit volume in cortical layer 1 for three different mice (Fig. 4e). Although there were slight variations among different mice and observation areas, substantial growth was observed, especially during the postnatal weeks 3–5. The system's capability of high-resolution deep-brain imaging enabled us to investigate the myelination processes even at layer 4 (Fig. 4f). Like cortical layer 1, we observed the emergence of new myelin segments (yellow arrowheads) as well as the persistence of the existing

segments (blue arrowheads). In fact, this is the first label-free observation of individual myelin segments at layer 4 through an intact skull. Notably, the image contrast of the myelin segments tended to increase over time (for example, see Fig. 4c). Since the lipid constituting the myelin is the main source of the intrinsic reflectance contrast, this observation implies the gradual thickening of the myelin. In addition to the myelin, blood vessels stretching laterally (green arrowheads in Fig. 4a) and axially (blue arrowheads in Fig. 4a) were also visible.

## Discussion

In summary, our 1.3-µm RMM system and image reconstruction algorithm enabled the monitoring of individual myelin segments in the cortical brain, without cranial surgery. This allowed us to investigate the myelination processes at the early developmental stages that are prone to surgical side effects. The use of the 1.3-µm wavelength source made a major contribution to the increase of imaging depth due to the substantial reduction of multiple scattering and aberration. This is in line with the earlier studies demonstrating the benefit of using the 1.3-µm wavelength source[6]. Specifically, we could barely reconstruct the myelin fiber right beneath the skull by the previous 900 nm system with the pupil-CLASS algorithm. On the contrary, we could image myelin segments at layers 4–5 with the 1.3-µm system even when the same pupil-CLASS algorithm was used.

In addition to the suppression of scattering and aberration by a longer wavelength source, we found that the conjugate-CLASS algorithm played a critical role. In the previous pupil-CLASS, the full-field image was divided into small subregions with sizes comparable to the isoplanatic patch (15 × 15 µm²) within which the aberration is identical. The fidelity of aberration correction is determined by the competition between the coherent addition of signals and the incoherent addition of multiple scattering noise, which favors a large isoplanatic patch. Therefore, a small isoplanatic patch size results in reduced fidelity of the aberration correction, especially when the skull thickness increases with aging. There is also degeneracy in the tilt and ambiguity in the defocus (Supplementary section 4.4), which can give rise to the lateral and axial shifts of reconstructed images in the individual patches. This makes it difficult to trace myelin segments in the volume and causes the blur of the MIP images in the pupil-CLASS (Supplementary Fig. 9). In contrast, the conjugate-CLASS algorithm could provide reliable aberration correction over the wide field of view, whose size is mainly determined by the computer memory. In our present study, this field of view was approximately 80 × 80 µm², i.e., ~30 times the isoplanatic patch (Supplementary Fig. 8).

It should be noted that the physical meaning of the aberration map in conjugate-CLASS differs from that of the pupil-CLASS. The aberration map of the conjugate-CLASS corresponds to the effective spatial phase retardation in the plane conjugate to the skull, while that of the pupil-CLASS is the angle-dependent phase retardation. Their direct comparison requires the conversion of the aberration map from one basis to the other (Supplementary section 4.3). The implementation of the conjugate-CLASS was possible because of the recording of the reflection matrix, which is a replica of the real optical system. Using the reflection matrix, one can freely choose the basis of illumination/detection from the pupil plane to any plane suitable for correcting the skull aberrations in post-processing. The present study acquired the reflection matrix by setting the objective focus to the cortical layers underneath the skull with the reference beam path matched to the focal plane. One can also fix the objective focus to the skull plane and scan only the time-gating window throughout the cortical layers to obtain reflection matrices for the volumetric image acquisition. This procedure can have potential benefits in that there is no need to scan the objective focus and convert the basis of the matrix. However, the signal collection efficiency can be substantially reduced with the increase of the target depth due to the defocus from the objective focal plane.

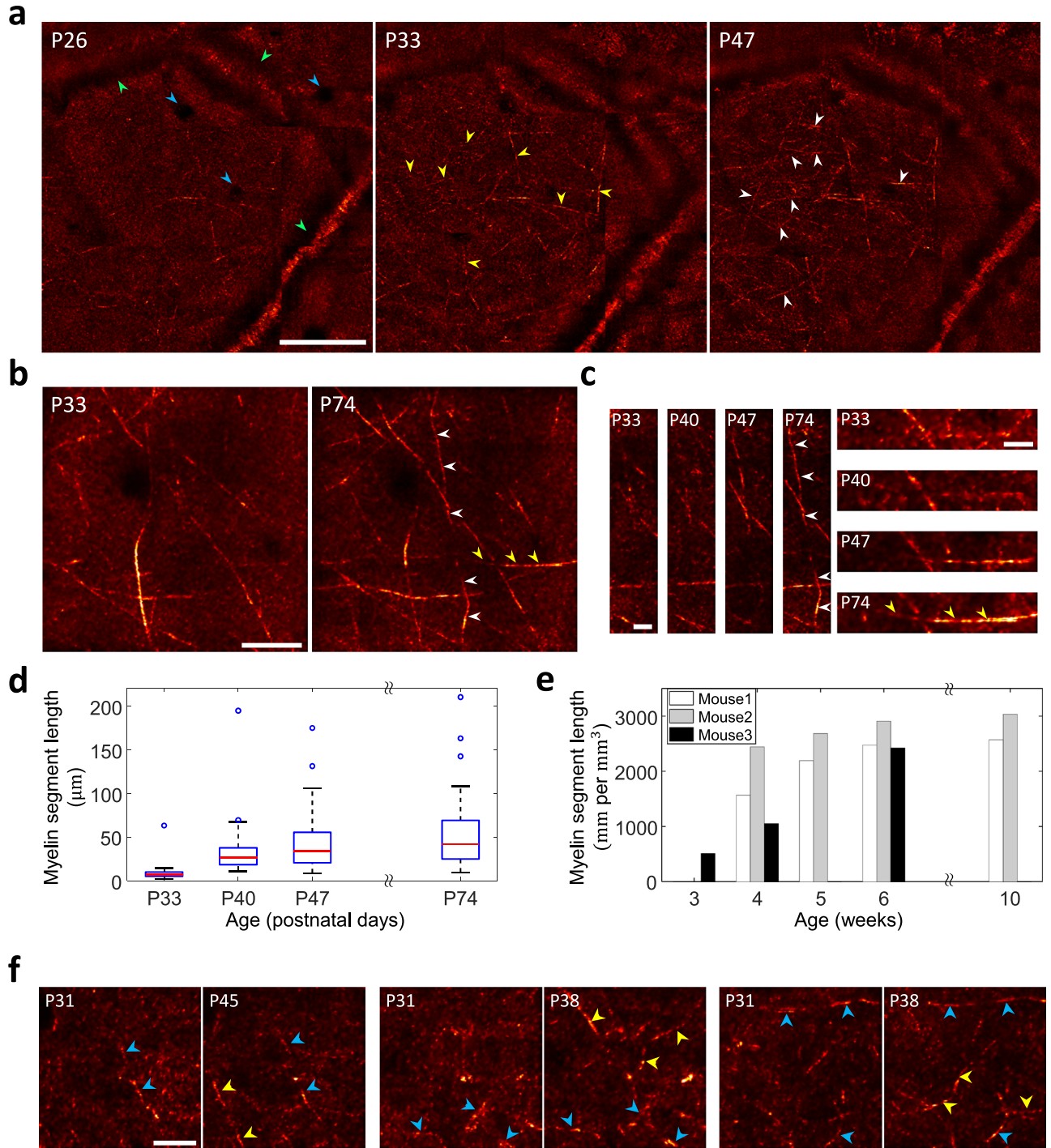

**Fig. 4 | Longitudinal observation of the myelination processes up to cortical layer 4 through the intact skull. a** Aberration-corrected images at cortical layer 1 of the same mouse recorded at P26, P33, and P47. Each image is an MIP across a 40-μm depth range. The yellow and white arrowheads mark the newly emerged myelin segments. Blood vessels oriented along the lateral and axial directions are marked with green and blue arrowheads, respectively. Scale bar: 100 μm. **b** Aberration-corrected images at cortical layer 1 of a different mouse from **a**, taken at P33 and P74. Each image is a MIP across a 20-μm depth range. Scale bar: 30 μm. **c** Zoomed-in images of the myelin segments marked with white and yellow arrowheads shown in **b**. Images are shown at P33 and P74 as well as at P40 and P47. Scale bars: 10 μm. **d** Box and whisker plots of myelin segment lengths measured in the images in

**b** with ages. Data represents the length of individual myelin segments in days P33 ($n = 61$), P40 ($n = 54$), P47 ($n = 47$), P74 ($n = 51$), where $n$ is the number of myelin segments. The red line represents the median value and open dots are outliers. The box boundaries show the upper and lower quartiles. Whiskers represent maximum and minimum values. **e** Total myelin segment length per unit volume derived from three different mice with age. **f** Aberration-corrected images on two different postnatal days at cortical layer 4. Three examples are shown. These individual pairs were MIPs over the depths of 405–435, 455–485, and 510–540 μm (from the left). The newly emerged myelin segments were marked with yellow arrowheads, whereas the remaining unchanged ones with age were marked with blue arrowheads. Scale bar: 20 μm.

Our system can serve as a potential platform for investigating the early development of learning and memory associated with myelination[21]. It can also be used for understanding myelin-related disorders, such as multiple sclerosis and leukodystrophies[22], and developing their treatment strategies. Our current system could conduct through-skull imaging of mice in the developing stages. Although there are variations depending on individual mice, our system could image cortex layer 1 for up to 10 weeks, close to adulthood. However, we could image mice about 7 weeks old in the case of cortex layer 4. We expect that a further increase in the source wavelength to 1700 nm will help to reach cortical layer 4 of the adult mice[7]. Technically, our system can be used as a wavefront sensing AO for nonlinear deep-brain imaging modalities, such as two- or three-photon microscopy, to recover their resolving power in the through-skull imaging[23]. This will facilitate longitudinal deep-brain imaging of many interesting neuronal structures, such as axons and dendrites, as well as non-neuronal structures, such as oligodendrocytes and microglia. Our work can also be extended to ultrasound imaging where the reflection matrix is routinely measured[24,25].

## Methods

### Computational conjugate adaptive optics algorithm

We developed a conjugate-CLASS algorithm that finds $\phi_{\text{in}}(\mathbf{w}_{\text{in}})$, $\phi_{\text{o}}(\mathbf{w}_{\text{o}})$, and $\widetilde{O}_M$ from $E(\mathbf{w}_{\text{o}};\mathbf{w}_{\text{in}})$ in Eq. (1). Here, we exploit the fact that the object spectrum $\widetilde{O}_M(k(\mathbf{w}_{\text{o}}+\mathbf{w}_{\text{in}})/z)$ in $E(\mathbf{w}_{\text{o}};\mathbf{w}_{\text{in}})$ is shift-invariant with respect to $\mathbf{w}_{\text{in}}$. In other words, the functional form of the object spectrum is maintained with the change in $\mathbf{w}_{\text{in}}$. The algorithm consists of two steps, first finding $\phi_{\text{in}}(\mathbf{w}_{\text{in}})$ and then $\phi_{\text{o}}(\mathbf{w}_{\text{o}})$. We first replace $\mathbf{w}_{\text{o}}$ with $\mathbf{W}=\mathbf{w}_{\text{o}}+\mathbf{w}_{\text{in}}$, which converts $E(\mathbf{w}_{\text{o}};\mathbf{w}_{\text{in}})$ into

$$E(\mathbf{W};\mathbf{w}_{\text{in}}) = -\frac{e^{2ikz}}{\lambda^2 z^2}e^{i\phi_{\text{o}}(\mathbf{W}-\mathbf{w}_{\text{in}})}\widetilde{O}_M(k\mathbf{W}/z)e^{i\phi_{\text{in}}(\mathbf{w}_{\text{in}})} + E_M(\mathbf{W};\mathbf{w}_{\text{in}}). \quad (2)$$

For each $\mathbf{w}_{\text{in}}$, we compute the angle of the inner product $\langle E(\mathbf{W};\mathbf{w}_{\text{in}})E^*(\mathbf{W};\mathbf{w}_{\text{in}}=0)\rangle_{\mathbf{W}}$ to obtain the first estimate of $\phi_{\text{in}}(\mathbf{w}_{\text{in}})$:

$$\phi_{\text{in}}^{(1)}(\mathbf{w}_{\text{in}}) = \phi_{\text{in}}(\mathbf{w}_{\text{in}}) + \delta\phi_{\text{in}}^{(1)}(\mathbf{w}_{\text{in}}). \quad (3)$$

Here, $\langle\rangle_{\mathbf{W}}$ indicates the summation of the terms in the bracket over all the possible $\mathbf{W}$. We set $\phi_{\text{in}}(\mathbf{w}_{\text{in}}=0)=0$ since only the relative phase matters. The error term $\delta\phi_{\text{in}}^{(1)}(\mathbf{w}_{\text{in}})$ arises due to the presence of $\phi_{\text{o}}(\mathbf{W}-\mathbf{w}_{\text{in}})$ and the multiple scattering term. We apply the correction of this first estimation by multiplying $e^{-i\phi_{\text{in}}^{(1)}(\mathbf{w}_{\text{in}})}$ to $E(\mathbf{W};\mathbf{w}_{\text{in}})$. Once this correction is in place, we move to the next step to correct $\phi_{\text{o}}(\mathbf{w}_{\text{o}})$.

The correction of $\phi_{\text{o}}(\mathbf{w}_{\text{o}})$ is similar to that of $\phi_{\text{in}}(\mathbf{w}_{\text{in}})$. We replace $\mathbf{w}_{\text{in}}$ with $\mathbf{W}=\mathbf{w}_{\text{o}}+\mathbf{w}_{\text{in}}$ and compute the angle of the inner product $\langle E(\mathbf{w}_{\text{o}};\mathbf{W})E^*(\mathbf{w}_{\text{o}}=0;\mathbf{W})\rangle_{\mathbf{W}}$ to find the approximate $\phi_{\text{o}}(\mathbf{w}_{\text{o}})$:

$$\phi_{\text{o}}^{(1)}(\mathbf{w}_{\text{o}}) = \phi_{\text{o}}(\mathbf{w}_{\text{o}}) + \delta\phi_{\text{o}}^{(1)}(\mathbf{w}_{\text{o}}). \quad (4)$$

After applying the output correction by multiplying $e^{-i\phi_{\text{o}}^{(1)}(\mathbf{w}_{\text{o}})}$ to $E(\mathbf{w}_{\text{o}};\mathbf{W})$, we go on an iteration for the successive input and output corrections. The iteration stops when the magnitudes of $\delta\phi_{\text{in}}^{(n)}(\mathbf{w}_{\text{in}})$ and $\phi_{\text{o}}^{(n)}(\mathbf{w}_{\text{o}})$ are smaller than a certain tolerance level after $n$ iterations. Then, the total corrected input and output aberrations are respectively given as

$$\phi_{\text{in}}^{\text{c}}(\mathbf{w}_{\text{in}}) = \sum_{j=1}^{n}\delta\phi_{\text{in}}^{(j)}(\mathbf{w}_{\text{in}}). \quad (5)$$

and

$$\phi_{\text{o}}^{\text{c}}(\mathbf{w}_{\text{o}}) = \sum_{j=1}^{n}\delta\phi_{\text{o}}^{(j)}(\mathbf{w}_{\text{o}}), \quad (6)$$

With the input and output aberration corrections in place, $E(\mathbf{w}_{\text{o}};\mathbf{w}_{\text{in}})$ is corrected to

$$E_{\text{c}}(\mathbf{w}_{\text{o}};\mathbf{w}_{\text{in}}) = e^{-i\phi_{\text{o}}^{\text{c}}(\mathbf{w}_{\text{o}})}E(\mathbf{w}_{\text{o}};\mathbf{w}_{\text{in}})e^{-i\phi_{\text{in}}^{\text{c}}(\mathbf{w}_{\text{in}})}. \quad (7)$$

We then replace $\mathbf{w}_{\text{o}}$ with $\mathbf{W}=\mathbf{w}_{\text{o}}+\mathbf{w}_{\text{in}}$ in $E_{\text{c}}(\mathbf{w}_{\text{o}};\mathbf{w}_{\text{in}})$:

$$E_{\text{c}}(\mathbf{W};\mathbf{w}_{\text{in}}) \cong -\left(e^{2ikz}/\lambda^2 z^2\right)\widetilde{O}_M(k\mathbf{W}/z) + E_M'(\mathbf{W};\mathbf{w}_{\text{in}}). \quad (8)$$

The summation of $E_{\text{c}}(\mathbf{W};\mathbf{w}_{\text{in}})$ with respect to $\mathbf{w}_{\text{in}}$ for $N$ orthogonal pixels leads to the object spectrum.

$$\sum_{\mathbf{w}_{\text{in}}}^{N}E_{\text{c}}(\mathbf{W};\mathbf{w}_{\text{in}}) \cong -\left(e^{2ikz}/\lambda^2 z^2\right)N\widetilde{O}_M(k\mathbf{W}/z). \quad (9)$$

The summation of $E_M'(\mathbf{W};\mathbf{w}_{\text{in}})$ is incoherent such that its magnitude grows with $\sqrt{N}$. Therefore, its contribution becomes smaller than the term with the object spectrum for a sufficiently large $N$. Taking the inverse Fourier transform of Eq. (9) provides the object function $O_M(\mathbf{r})$. See the full description of the conjugate-CLASS algorithm in Supplementary sections 2 and 3.

### Animal preparation for the longitudinal through-skull imaging

Three- to 4-week-old C57BL/6 mice (P21-P28; body weight: 9–14 g) were anesthetized with isoflurane (2% in oxygen for induction, and 1.0–1.5% in oxygen for surgery to maintain a breathing frequency of ~1 Hz). The body temperature was kept at 37–38 °C by a heat blanket, and the eyes were protected with an eye ointment during the surgery and imaging. Dexamethasone (2 mg/kg) was administrated via subcutaneous injection to minimize swelling at the surgery site. The hair and scalp were removed to expose the bregma and lambda, and both parietal plates of the skull. The connective tissue remaining on the skull was gently removed with sterile saline. After the saline that covered the skull was completely removed, a sterile coverslip of 5-mm diameter was attached to the center of the parietal bone using an ultraviolet-curable glue. A custom-made metal plate was attached to the skull with cyanoacrylate for head fixation during the in vivo imaging, and the exposed part of the skull was covered with dental cement. Finally, the window was covered by a biocompatible silicone sealant (Kwik-Cast, World Precision Instruments) for protection until the imaging was performed. For imaging, the mice were anesthetized with isoflurane (1.2–1.5% in oxygen to maintain a breathing frequency of ~1.5 Hz) and placed on a 3D motorized stage heated by a heat blanket at 37–38 °C. The mice were weighed each time they were imaged during the experiment. Mice were maintained in temperature (20–22 °C) and humidity (50–55%) controlled facilities with 12-h light/12-h dark cycles. All the animal-related procedures were approved by the Korea University Institutional Animal Care and Use Committee (KUIACUC-2019-0024).

### Measuring myelin segments

To measure the length of myelin segments in cortical layer 1 (Fig. 4d, e), we analyzed the aberration-corrected images from the cortical surface to a depth of ~100 μm of all the mice investigated in this study. The aberration-corrected images acquired using MATLAB were converted to TIFF images. For measuring the myelin segment lengths, these images were transferred to ImageJ. The image brightness and contrast levels were adjusted for clarity, and the size of the image was defined with "Set Scale." Myelin sheaths within the field of view were drawn by a few investigators, and the lengths of the myelin segments were measured automatically.

### Reporting summary

Further information on research design is available in the Nature Portfolio Reporting Summary linked to this article.

## Data availability
The datasets acquired for this study are available from the corresponding authors upon request. The size of the datasets is too large to upload to the publicly accessible repository.

## Code availability
The codes developed in this study are available from the authors upon request.

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

## Acknowledgements
This work is supported by the Institute for Basic Science (IBS-R023-D1) (Y.K., J.H.H., S.K., H.L., Y.J., S.Y., and W.C.) and the National Research Foundation (NRF) of Korea grant (NRF-2020R1A2C3009309) (K.H.K.).

## Author contributions
Y.K. and S.Y. designed and constructed an experimental setup along with H.L. J.H.H. designed mouse brain imaging sessions along with K.H.K. and undertook animal preparations. S.K. and S.Y. developed the image reconstruction algorithm. Y.K. and J.H.H. conducted experiments, and they performed data analysis and image processing along with S.K. and Y.J. Y.K., J.H.H., S.K., S.Y., and W.C. prepared the manuscript, and all authors contributed to finalizing the manuscript. W.C. supervised the project.

## Competing interests
The authors declare no competing interests.
