## [Peer Review File · Nature Communications]

Computational conjugate adaptive optics microscopy for longitudinal through-skull imaging of cortical myelinREVIEWER COMMENTS

Reviewer #1 (Remarks to the Author):

In this work, the authors investigate a reflection matrix approach of label-free microscopy that allows to overcome aberrations induced by a mice skull and image cortical myelin. Compared to the previous work of the same group (Ref. 11 -> Yoon et al., Nature Comm, 2019) that investigated the same problem, they make evolve their algorithm (CLASS) to analyze the reflection matrix not in the k-space but in a plane conjugated with the aberration layer, here the skull. Thereby, they numerically mimic the principle of conjugate adaptive optics. The experimental set up is equivalent to Ref. 11, only the light source has been modified to be in the 1.3 μm wavelength instead of the standard therapeutic window in biological tissues (800-850 nm). This choice is made to reduce scattering and aberration of biological tissues. A first experimental proof-of-concept is provided by introducing a resolution target behind the skull. They show a clear improvement compared with the original CLASS algorithm since the new basis allows to maximize the size of isoplanatic patches. The authors then perform through-skull deep-brain imaging up to cortical layer 4 of a five-week-old mouse.

In overall, this paper is of good quality, well written, concise and clear. This work is useful both for researchers working in the field of wave-front shaping and those more generally motivated by the difficult and long-standing challenge of imaging in/through complex media. I think it perfectly fits with the scope of Nature Communications. One open question is the novelty of this work with regards to their own previous papers also published in Nature Communications [11,17]. For non-specialists, this work may be seen as only incremental, especially with respect to Ref.11, except that the experimental in-vivo demonstration is here more spectacular. Nevertheless, I think the authors did not discuss the in-vivo issues enough, in particular the decorrelation of tissues that can be very rapid in living tissues (<100 ms) and that the authors seem to neglect in their analysis. Yet, the measurement of the reflection matrix for a single focal plane is already of 2 s. Another issue is the exact nature of aberrations in their experiment. I suspect that a large contribution of aberrations is due to a strong defocus (see Fig.2d) due to the imaging system. This defocus is induced by the mismatch between the coherence plane (controlled by the reference arm) that is used to scan in depth the sample and the fixed position of the MO focal plane in the sample arm. As it is deterministic, this defocus could have been compensated beforehand in order to focus only on the skull-induced aberrations and make a fairer comparison with the state-of-the-art. In that respect, the provided signal enhancement and PSF narrowing could be revised.

Below, I give more specific comments on the previous points and others that need to be addressed by the authors in order to improve the quality of their manuscript:

1/ In the resolution target experiment, the authors indicate that *"it took 2.6 s to scan the focus across a field of illumination of $80 \mu\text{m}^2 \times 80 \mu\text{m}^2$ with a scanning step size of the diffraction limit"*. This measurement time does not allow real-time imaging and is of course much longer for the 3D (longitudinal) imaging of the cortical myelin. Could the authors provide the overall measurement time for each movie shown in Supplementary materials as longitudinal imaging is one of the main achievement in this work?

2/ Living tissues exhibit a decorrelation characteristic time ranging from 50 ms to 2.5 s depending on the level of immobilization [M. Jang et al., Biomed. Opt. Express 6, 72 (2014)]. The authors should at least discuss this point in the manuscript. The reflection matrix approach makes the implicit hypothesis of a static medium which is everything but true in-vivo. To see if this hypothesis is true in their experiment, the authors should provide a measurement of this decorrelation time. If not, they should

assess to which extent the medium movement can affect their imaging process and how the new CC-AO algorithm can deal with it.

3/ In Fig.1, the authors show a PSF before and after correction but the authors did not explain how they record it. If this PSF is deduced from the reflection matrix, it is not the “true” PSF but an apparent PSF measured in reflection. It then corresponds to a convolution between the incoherent input and output PSFs [see W. Lambert et al., PRX, 2020]. The way this PSF is measured should be clearly stated in the manuscript.

4/ Later, in the manuscript, the authors claim that: “The CC-AO algorithm converted the confocal image's substantially broadened point spread function (PSF) (approximately 30 μm in FWHM) into a near-diffraction-limited PSF (0.66 μm in FWHM) (Fig.1l).” Actually, this is not strictly true since a confocal image exhibits theoretically a diffraction-limited resolution close to $\lambda/4NA$: The confocal imaging PSF is actually equal to the product between the input and output PSF [see, for instance, A. Sentenac and J. Mertz, JOSA A, 2018 or V. Barolle et al., Opt. Exp., 2021]. I guess the resolution is here rather limited by the spatial sampling of the scanning scheme which is equal to $\lambda/2NA$. The claim that the resolution is nearly diffraction-limited is thus not strictly true. The same remark holds for the claim made in the abstract.

5/ In the in-vivo experiment, the authors divide the area of individual subregions to be 64 \times 64 μm^2 to cope with the lateral variation of aberrations induced by the skull. The authors should indicate how they fix this size. What are the physical phenomena or mathematical reasons that dictate this choice? On the one hand, I guess the best option would be to estimate aberration phase laws over the smallest possible isoplanatic patch to grasp the high-order spatial variations of the aberration phase laws. On the other hand, the isoplanatic patch should be large enough to converge towards a satisfying aberration phase law. A discussion is therefore lacking about the choice made for the size of these subregions.

6/ In Fig.2, the aberration maps mainly correspond to a defocus induced by the mismatch between the coherence plane (whose position is controlled by the reference arm) and the focal plane of the microscope objective. Hence, a major part of the aberrations is not due to the skull but to the imperfections of the imaging system. I think this should be clearly stated in the manuscript. When the PSF width or Strehl ratio are estimated, it would be fair to provide the relative proportions between defocus and skull aberrations.

7/ This defocus that grows with z also explains why “*We observed that the signal enhancement after the aberration correction grew exponentially with depth*”. This signal enhancement is therefore somehow artificial since it could have been *a priori* compensated in their algorithm. The same remark holds for the following sentence: “*The PSF width was broadened by 44 times relative to the diffraction limit at cortical layer 4, supporting the difficulty in reaching cortical layer 4 via the intact skull.*”. I think there might be an overstatement here. State-of-the-art 3D imaging techniques, such as confocal microscopy or OCT, can compensate for such a defocus. Hence I think the impact of defocus and skull-aberrations should be discriminated.

8/ Related to this question of defocus, would that be a better option to measure directly the reflection matrix directly in the (w_{in}, w_{out}) basis? In that case, the authors could have used the CLASS algorithm directly on the recorded reflection matrix. I think it would be nice to discuss this option and argue about its advantages/defaults compared to the authors' choice to measure the reflection matrix in a plane conjugated with the focal plane of the microscope objective.

9/ Do the aberration maps displayed in Fig.2d correspond to input or output aberration phase laws? Could the authors show a comparison between input and output aberration phase laws to see whether

they differ or not. Normally they should not because of spatial reciprocity but I guess this property is also extremely sensitive to alignment issues and other imperfections of the imaging system.

10/ When discussing the merit of their CC-AO algorithm with respect to CLASS, the authors mention that in the original CLASS *"There is also degeneracy in the tilt and defocus in the pupil AO, which can give rise to the lateral and axial shifts of reconstructed images in the individual patches"*. I'm not sure to understand what they mean by "degeneracy". Can the authors be more explicit?

I congratulate the authors for their work and hope they will find my comments useful and relevant.

Alexandre Aubry

Reviewer #2 (Remarks to the Author):

The authors present a novel development of their optical coherence microscopy-related adaptive optics method designed to amplify the signal from the back-reflected single-scattered waves inside biological media. Originally described as CASS (Kang et al., Nat Phot 2015), it allows one to increase the signal from waves back-scattered from the target inside a scattering medium, while decreasing the contribution of the multiple-scattered light at the same time through the use of time gating and coherent summation of the back-reflected waves. Further developments (termed CLASS, Kang et al., Nat Comm 2017) allowed the authors to correct aberrations separately for the light scattered on the way to and away from the imaging plane. Having already demonstrated its utility in imaging myelin fibers through intact skulls in a mouse (Yoon et al., Nat Comm 2020), in the present study the authors expand this technique through the use of computational conjugate adaptive optics to increase the field of view after aberration correction. As the majority of aberrations comes from the intact skull, correcting the aberrations in the skull plane allowed authors to image myelin fibers deeper inside the mouse cortex with larger FOV and better signal-to-noise ratio. Furthermore, the computational propagation to the conjugate plane helped avoid time-consuming alignment of the setup for hardware conjugate AO. The authors also changed the wavelength from 900 nm to 1300 nm to reduce scattering.

I found this study to be an interesting development of the imaging methods based on single-scattered waves. However, I would like the authors to address the following issues before it can be published in Nature Communications:

- The data processing is insufficiently described in the paper, even if it was described in the previous studies. Why did the authors change notation vs. their previous papers? Data analysis steps with a corresponding data analysis pipeline should be demonstrated in the Supplementary methods.
- What are the conditions for convergence of their method? How do authors show that maximum intensity in the imaging plane results in a correct estimation of the aberration field? Is sparsity of the sample and strong back-scattering of the target structures an absolute prerequisite for this method to converge? Would this limit the applicability for other types of samples?
- Prior related work in ultrasound not referenced or even mentioned.
- The authors compare their method to optical coherence microscopy in the time domain. However, in many applications OCM/OCT is used in frequency domain due to the increased SNR. How would their method compare to the frequency-domain OCM? What would be the benefits in terms of SNR and imaging depth? How much faster/slower would the imaging be?
- Inconsistent method nomenclature. Why not pupil CLASS vs. conjugate CLASS as method names? Authors use these, RMM and CC-AO interchangeably, which is confusing, even if they reference different parts of their methods.
- From the text and mathematical derivations it follows that the skull plane doesn't need to be physically conjugate but in Fig. 1c it is. Fig. should reflect it clearly.
- Not all components in Fig 1a are labelled (pinhole, beam splitter in front of PD). A list of lenses/components used would be useful for reproducibility.
- Please check the language. For example: use of "ingenious" seems misplaced.

Reviewer #3 (Remarks to the Author):

This group has significantly improved their original CLASS system and algorithm which originally performed computational adaptive optics at 900 nm. First, they built a system at 1300 nm to take advantage of reduced scattering in the brain and skull, and second, they applied their algorithm in the plane of the skull rather than the pupil plane of the sample. As the major aberrations occur in the skull, according to their claim, this gives them a larger isoplanatic patch and simplifies image

reconstruction. Myelinated axons were imaged in layer 4 in young mice over time with apparently high resolution.

I have the following criticisms which should be addressed in the manuscript text:

- 1) Younger mice have thinner skulls which are less scattering and also less myelin making it easier to see deeper. The mice imaged here are not yet adults and are still developing myelination. How the system will perform in adult or aged (years old) mice is not clear.
- 2) The SHG inset in Fig. 2A has a depth axis that appears inconsistent with the rest of the figure.
- 3) They talk about tracing myelin ('In summary, our 1.3- μm RMM system and image reconstruction algorithm enabled the tracing of individual myelin segments in the cortical brain'), but I feel they are just monitoring individual myelinated segments, not really tracing them.
- 4) Recent 1700 nm OCM work should be cited; Zhu, J., Freitas, H.R., Maezawa, I. et al. 1700 nm optical coherence microscopy enables minimally invasive, label-free, in vivo optical biopsy deep in the mouse brain. *Light Sci Appl* 10, 145 (2021). <https://doi.org/10.1038/s41377-021-00586-7>
- 5) The authors state: "genetic labeling can visualize only a small fraction of mature myelination owing to its partial expression in a small population of oligodendrocytes"- I believe that this partial expression might be by design, not by necessity, so that individual myelin axons can be traced and seen.
- 6) They cite Ref. 10 for the inflammatory response of skull thinning and opening but not earlier work which has shown skull thinning with glass reinforcement to be significantly less inflammatory than skull opening: Drew, P., Shih, A., Driscoll, J. et al. Chronic optical access through a polished and reinforced thinned skull. *Nat Methods* 7, 981–984 (2010). <https://doi.org/10.1038/nmeth.1530>. Moreover, a more accurate statement of Ref. 10 is that surgical skill is an important determinant of outcome with skull thinning, with inadvertent mechanical trauma causing CNS disruptions (even this is still a good motivation for the approach presented here though)
- 7) Are the conjugate plane aberration maps in Fig. 2D covering/obscuring some of the sub-region aberration maps? Please shift to the side. Also does 'conjugate plane' mentioned in the caption refer to the conjugate plane of the skull? Please explicitly state. Why is the scale bar different than the others in the top map in panel d? Also why are these aberration maps 'representative'-I thought with the conjugate plane algorithm there should be only one aberration map in the skull plane.
- 8) Please explain why the reflection matrix depends so strongly on the illuminated position in Fig. 1d? This suggests to me that system aberrations may be dominant, as I don't see why skull aberration should be radially symmetric with respect to the field of view.
- 9) Was any effort made to measure or correct system (not brain) aberrations independently of CLASS algorithm?
- 10) Please make clear that Fig. 1c is not the actual experimental scheme, and reflects a hypothetical setup.
- 11) Also shouldn't both panels d and e in Fig. 1 have an amplitude and a phase? If this is the case, please explain that the phase is not being shown.
- 12) Pg 2: "observed the emergence of new myelin segments in cortical layer 4 with aging."-> I would say 'with development' instead of 'with aging'
- 13) They state that they are "exploiting the shift-invariance of OM with respect to w_{in} "
 - a. Is it also shift invariant with respect to w_{out} ?
 - b. Does shift-invariant mean that it does not depend of w_{in} ? If true please rephrase in a simpler way.
 - c. Can you restate this as a condition on OM (amplitude reflectance of object)?
- 14) They state 'The aberration-corrected images were acquired at a depth interval of 3 μm from the cortical surface to a depth of $\sim 100 \mu\text{m}$ in cortical layer 1 of all the mice'-was imaging not performed to layer IV in some mice?
- 15) Their algorithm remains conceptually similar to CLASS (as I understand they apply a numerical propagation to a different plane prior to applying their algorithm). It appears that most of the improvement is coming from the longer wavelength used in this study. The comparison between 900 nm and 1.3 microns merits more attention.
- 16) In the Supplement Section 2.3 it is not at all clear if they have made any modifications to their iterative algorithm relative to prior CLASS works-please explicitly state if there are differences, or cite the prior work if not.
- 17) I don't understand why ϕ_{in} and ϕ_{out} don't have some relationship and are treated as totally independent-Aren't these double pass aberrations? Are all aberration maps (e.g. 2d) related to ϕ_{in} ? What about ϕ_{out} ?

Reviewer #1

In this work, the authors investigate a reflection matrix approach of label-free microscopy that allows to overcome aberrations induced by a mice skull and image cortical myelin. Compared to the previous work of the same group (Ref. 11 -> Yoon et al., Nature Comm, 2019) that investigated the same problem, they make evolve their algorithm (CLASS) to analyze the reflection matrix not in the k-space but in a plane conjugated with the aberration layer, here the skull. Thereby, they numerically mimic the principle of conjugate adaptive optics. The experimental setup is equivalent to Ref. 11, only the light source has been modified to be in the 1.3 μm wavelength instead of the standard therapeutic window in biological tissues (800-850 nm). This choice is made to reduce scattering and aberration of biological tissues. A first experimental proof-of-concept is provided by introducing a resolution target behind the skull. They show a clear improvement compared with the original CLASS algorithm since the new basis allows to maximize the size of isoplanatic patches. The authors then perform through-skull deep-brain imaging up to cortical layer 4 of a five-week-old mouse.

In overall, this paper is of good quality, well written, concise and clear. This work is useful both for researchers working in the field of wave-front shaping and those more generally motivated by the difficult and long-standing challenge of imaging in/through complex media. I think it perfectly fits with the scope of Nature Communications.

We appreciate the reviewer for carefully reviewing our manuscript and acknowledging the importance of our work. The reviewer's critical comments helped us to improve the scientific integrity of the manuscript.

One open question is the novelty of this work with regards to their own previous papers also published in Nature Communications [11,17]. For non-specialists, this work may be seen as only incremental, especially with respect to Ref.11, except that the experimental in-vivo demonstration is here more spectacular.

We agree with the reviewer's concern in that the novelty of the present work should be better emphasized for non-specialists. As the reviewer acknowledged, our work presented a spectacular demonstration of *in vivo* through-skull imaging. Specifically, we could visualize fine myelin segments up to cortical layer 4 for the first time in the label-free through-skull imaging and realize the long-term longitudinal observation of the growth of the myelination process. This is substantial progress with respect to our previous work [Yoon, S. *et al.*, *Nat Commun.* 11, 5721 (2020)] relying on pupil-CLASS at the source wavelength of 0.9 μm , in which we could reach only up to cortical layer 1.

We made two noteworthy advances with respect to our earlier work, one in the hardware and the other in the software. We constructed a high-speed reflection matrix microscope using a light source with a wavelength of 1.3 μm , instead of 0.9 μm , which played a critical role in reducing tissue scattering and aberration. Furthermore, we developed a computational conjugate adaptive optics algorithm to optimally compensate for the skull aberrations. In the case of the pupil-CLASS developed in our earlier study, the isoplanatic size was so small that the correction algorithm should be applied repeatedly over numerous small patches constituting a large field of view (FOV). This is computationally demanding and, more importantly, the

44 image intensity is not uniform even within small patches. Therefore, the merged full field image
was fragmented, making it difficult to visualize long myelin segments extending across
multiple isoplanatic patches. The newly developed conjugate-CLASS algorithm enabled us to
substantially enlarge the isoplanatic patch such that uniform image quality is acquired over a
large FOV (see the direct comparison between pupil-CLASS and conjugate-CLASS in
Supplementary Fig. 8). Note that we renamed the term CC-AO as conjugate-CLASS following
the reviewer #2's suggestion.

The new algorithm realizes the confirmed benefits of conjugate AO over the pupil AO in the
context of the computational AO. In comparison with the previous hardware-based conjugate
AOs, this newly developed algorithm is flexible in the choice of the conjugate plane in the
post-processing steps and can remove much more complex aberrations due to the use of
computational optimization.

We added the following sentence to the introduction to emphasize the benefit of the newly
developed algorithm.

“In comparison with the previously developed pupil-CLASS^{13,16}, the conjugate-CLASS
substantially enlarges the isoplanatic patch size, thereby enhancing the image reconstruction
fidelity with reduced computational cost.”

*Nevertheless, I think the authors did not discuss the in- vivo issues enough, in particular the*
*decorrelation of tissues that can be very rapid in living tissues (<100 ms) and that the authors*
*seem to neglect in their analysis. Yet, the measurement of the reflection matrix for a single*
*focal plane is already of 2 s.*

This is an excellent question. A brief answer to this question is that a special mounting device
ensures long decorrelation times (> 30 s) enough to conduct *in vivo* imaging of live animals.
We will discuss this issue in detail in response to the reviewer's second comment.

*Another issue is the exact nature of aberrations in their experiment. I suspect that a large*
*contribution of aberrations is due to a strong defocus (see Fig. 2d) due to the imaging system.*
*This defocus is induced by the mismatch between the coherence plane (controlled by the*
*reference arm) that is used to scan in depth the sample and the fixed position of the MO*
*focal plane in the sample arm. As it is deterministic, this defocus could have been*
*compensated beforehand in order to focus only on the skull-induced aberrations and make*
*a fairer comparison with the state-of-the-art. In that respect, the provided signal*
*enhancement and PSF narrowing could be revised.*

We understand the reviewer's concern about the probable existence of strong defocus, judging
from the appearance of the defocus-like phase patterns in aberration maps. However, the
aberration map obtained in conjugate-CLASS corresponds to the spatial phase retardations
induced by the skull in the conjugate plane, not the angle-dependent phase retardation in the
pupil plane. Therefore, the effect of the aberration patterns on the PSF is different in the
conjugate AO. As we shall discuss in detail in response to comments #6 and #7, the degree of
defocus was negligible, and the correction of the fine aberration patterns was mainly
responsible for the recovery of the myelin images.

*Below, I give more specific comments on the previous points and others that need to be*
*addressed by the authors in order to improve the quality of their manuscript:*

***1/ In the resolution target experiment, the authors indicate that “it took 2.6 s to scan the***
***focus across a field of illumination of $80 \times 80 \mu\text{m}^2$ with a scanning step size of the diffraction***
***limit”. This measurement time does not allow real-time imaging and is of course much***
***longer for the 3D (longitudinal) imaging of the cortical myelin. Could the authors provide***
***the overall measurement time for each movie shown in Supplementary materials as***
***longitudinal imaging is one of the main achievements in this work?***

The acquisition time for each depth depends on the size of the region of interest (ROI). It takes
2.6 s for a ROI of $80 \times 80 \mu\text{m}^2$ and 10.4 s for $160 \times 160 \mu\text{m}^2$. The data acquisition time for
the 3D imaging depends on both ROI and the number of depth acquisitions. The total
acquisition time for each Supplementary Movie is given below.

Supplementary Movie 1: $160 \times 160 \mu\text{m}^2 \times 141$ layers, 1466.4 s (24.4 min)

Supplementary Movie 2: $160 \times 160 \mu\text{m}^2 \times 146$ layers, 1518.4 s (25.3 min)

Supplementary Movie 3: $160 \times 160 \mu\text{m}^2 \times 159$ layers, 1653.6 s (27.6 min)

This volumetric image acquisition speed is good enough for monitoring the development of
the myelination process occurring on a week-to-week time scale.

The through-skull imaging capability of our conjugate-CLASS comes from the acquisition of
a reflection matrix, which requires the camera recording over a wide field of detection (FOD)
for each focus scanning over the ROI. Therefore, the image acquisition speed is slower than
that relying on the integral detection by a photodetector. However, we can speed up the image
acquisition for a smaller ROI and FOD. For example, we can measure a reflection matrix for
the ROI of $40 \times 40 \mu\text{m}^2$ in less than a second.

We added the details of the data acquisition time to Supplementary section 1.2.

***2/ Living tissues exhibit a decorrelation characteristic time ranging from 50 ms to 2.5 s***
***depending on the level of immobilization [M. Jang et al., Biomed. Opt. Express 6, 72 (2014)].***
***The authors should at least discuss this point in the manuscript. The reflection matrix***
***approach makes the implicit hypothesis of a static medium which is everything but true in-***
***vivo. To see if this hypothesis is true in their experiment, the authors should provide a***
***measurement of this decorrelation time. If not, they should assess to which extent the***
***medium movement can affect their imaging process and how the new CC-AO algorithm can***
***deal with it.***

In our animal preparation procedure, a coverslip of 5 mm diameter was attached to the center
of the parietal bone using an ultraviolet-curable glue. A custom-made metal plate was attached
to the skull with cyanoacrylate for head fixation during the *in vivo* imaging, and the exposed
part of the skull was covered with dental cement. Therefore, our system can be considered as
a ‘significantly immobilized’ situation in the reference paper [Jang, M. *et al.*, Biomed. Opt.
Express 6, 72 (2015)].

Following the reviewer’s suggestion, we measured the decorrelation time in the *in vivo* imaging.
The acquisition time for a single-depth reflection matrix is as long as 10.4 s in our experiment.
Therefore, we measured the image decorrelation for 30 s. Specifically, we illuminated the
focused beam at a depth of 70 μm beneath the dura and acquired a series of complex-field
images over the field of detection of $75 \times 75 \mu\text{m}^2$ at a frame rate of 2,000 Hz.

We calculated the correlation r defined by the formula,

$$130 \quad r = \frac{\sum_m \sum_n (A_{m n} - \bar{A})(B_{m n} - \bar{B})}{\sqrt{(\sum_m \sum_n (A_{m n} - \bar{A})^2)(\sum_m \sum_n (B_{m n} - \bar{B})^2)}}.$$

Here A and B corresponds to the $I(t_0)$ and $I(t_0 + \tau)$, respectively, and \bar{A} and \bar{B} are their
respective mean values. The m and n refer to the m^{th} and n^{th} pixels, respectively, in each
image. This formula is the same as the function $g_2(\tau)$ introduced in the reference paper.
Figure R1 shows that a high correlation is maintained during the time of 30 s with a minimum
correlation value of 0.78. The regular peaks in the plot were due to the breathing of the mouse.
The correlation value remained higher than 0.85 for the time duration of 10.4 s, which is the
single-depth recording time for $160 \mu\text{m}^2 \times 160 \mu\text{m}^2$. This result supports that the tissue make
little effect on our in vivo imaging.

**Figure R1. Image correlation with time.** The correlation r was calculated from a set of time-gated
images acquired for 30 s while the focused illumination was parked at a depth of $70 \mu\text{m}$ beneath the
dura. The correlation value stayed higher than 0.78 over the entire measured time span.

We added the following sentence to the revised manuscript and added this new result to
Supplementary section 1.3.

“The decorrelation time due to the motion of the living mice was much longer than the
recording time of each reflection matrix.”

*3/ In Fig.1, the authors show a PSF before and after correction but the authors did not*
*explain how they record it. If this PSF is deduced from the reflection matrix, it is not the*
*“true” PSF but an apparent PSF measured in reflection. It then corresponds to a*
*convolution between the incoherent input and output PSFs [see W. Lambert et al., PRX,*
*2020]. The way this PSF is measured should be clearly stated in the manuscript.*

The PSF shown in Fig.1 was acquired by the ensemble average of the intensity PSFs before
and after the aberration correction from the reflection matrix. Therefore, the presented PSF
corresponds to the average intensity profile of the convolution between the input and output
PSFs, as the reviewer mentioned. To clarify this, we added the following sentence to the figure
caption.

“They correspond to the convolution of input and output PSFs, thereby accounting for the
roundtrip aberrations.”

We presented this apparent PSF as it fully represents the effect of aberrations on the reflectance
mode imaging. The PSF by the one-way aberration can be deduced from either input or output
aberration map.

*4/ Later, in the manuscript, the authors claim that: “The CC-AO algorithm converted the*
*confocal image's substantially broadened point spread function (PSF) (approximately 30*
*μm in FWHM) into a near-diffraction-limited PSF (0.66 μm in FWHM) (Fig.11).” Actually,*
*this is not strictly true since a confocal image exhibits theoretically a diffraction-limited*
*resolution close to $\lambda/4NA$: The confocal imaging PSF is actually equal to the product*
*between the input and output PSF [see, for instance, A. Sentenac and J. Mertz, JOSA A,*
*2018 or V. Barolle et al., Opt. Exp., 2021]. I guess the resolution is here rather limited by the*
*spatial sampling of the scanning scheme which is equal to $\lambda/2NA$. The claim that the*
*resolution is nearly diffraction-limited is thus not strictly true. The same remark holds for*
*the claim made in the abstract.*

As the reviewer pointed out, the diffraction-limited resolution of the confocal imaging is close
to $\lambda/4NA$. In fact, our system's diffraction limit is also close to $\lambda/4NA$ even though the
reflection matrix is acquired with $\lambda/2NA$ intervals of the focus scanning. This is because the
spatial frequency coverage is enlarged in the image reconstruction step. The coherent addition
based on aperture synthesis results in the doubling of the spatial-frequency bandwidth, thereby
enhancing the resolving power by a factor of two (Supplementary Information section II in S.
Kang et al., Nat. Photonics 9, 253 (2015)).

Even though our system can support $\lambda/4NA$ resolution, this is seldom achieved in real practice
partly because the amplitude transfer function is attenuated at high spatial frequencies and
partly because the biological target's object spectrum doesn't fill uniformly over the entire
bandwidth in the reflectance imaging in most cases. For this reason, we compared the
experimentally achieved resolution with the conventional diffraction limit of $\lambda/2NA$, not the
system's diffraction limit. Since this can cause confusion, we removed the term 'near-
diffraction-limited' in the revised manuscript.

*5/ In the in-vivo experiment, the authors divide the area of individual subregions to be 64*
*× 64 μm² to cope with the lateral variation of aberrations induced by the skull. The authors*
*should indicate how they fix this size. What are the physical phenomena or mathematical*
*reasons that dictate this choice? On the one hand, I guess the best option would be to estimate*
*aberration phase laws over the smallest possible isoplanatic patch to grasp the high-order*
*spatial variations of the aberration phase laws. On the other hand, the isoplanatic patch*
*should be large enough to converge towards a satisfying aberration phase law. A discussion*
*is therefore lacking about the choice made for the size of these subregions.*

The major advantage of the conjugate AO over pupil AO in the through-skull imaging is its
larger isoplanatic patch size. For the same experimental data, we found that the isoplanatic size
of the pupil-CLASS was around $15 \times 15 \mu\text{m}^2$ while that of our conjugate-CLASS was

around $80 \times 80 \mu\text{m}^2$ (Supplementary Fig. 8). Therefore, the size of each subregion to be
 analyzed can be as large as $80 \times 80 \mu\text{m}^2$. Since the recorded ROI was $160 \times 160 \mu\text{m}^2$, we
 can divide the entire ROI into 4×4 subregions with a 50 % overlapping ratio for optimal
 visibility. In this case, the computation time for reconstructing the entire ROI was 2,386 s. Note
 that the computation time is spent on the post-processing and, thus, does not affect to the in-
 vivo imaging. In the actual data analysis, we chose the subregion size of $64 \times 64 \mu\text{m}^2$ and
 divided the entire ROI into 5×5 subregions, which resulted in a reduced computation time of
 1557 s. In fact, as the patch size decreases and, thus, the number of patches increases, the total
 computation time gets shorter because the computation time for each patch scales with a power
 of 2 with respect to the area of the patch as shown in Fig. R2. Since we processed the data over
 140 to 170 depths in the volumetric image, this difference in computation time is a critical
 factor. We didn't reduce the subregion size to smaller than $64 \times 64 \mu\text{m}^2$ to avoid image
 fragmentation in merging multiple images.

We added this discussion to Supplementary section 1.4.

**Figure R2. Computation time depending on the analysis patch area.** Data points in the figure
 correspond to $40 \times 40 \mu\text{m}^2$, $46 \times 46 \mu\text{m}^2$, $52 \times 52 \mu\text{m}^2$, $64 \times 64 \mu\text{m}^2$, $80 \times 80 \mu\text{m}^2$, and
 $105 \times 105 \mu\text{m}^2$. A computer used for this analysis is equipped with i9-12900KS CPU and NVIDIA
 RTX A6000 GPU. Note that the computation time for the full field of view is given by the multiplication
 of the computation time shown here to the number of patches. The data was well-fitted with the second-
 degree polynomial function.

*6/ In Fig.2, the aberration maps mainly correspond to a defocus induced by the mismatch*
 *between the coherence plane (whose position is controlled by the reference arm) and the*
 *focal plane of the microscope objective. Hence, a major part of the aberrations is not due to*
 *the skull but to the imperfections of the imaging system. I think this should be clearly stated*
 *in the manuscript. When the PSF width or Strehl ratio are estimated, it would be fair to*
 *provide the relative proportions between defocus and skull aberrations.*

We assume that the reviewer's reasoning is based on the appearance of the defocus-like phase
 patterns in aberration maps. However, it is not these slowly varying phase patterns but the fine
 specular phase patterns that contribute mainly to the reconstruction of fine myelin segments.
 To support this claim, we conducted the Zernike decomposition of each aberration map and

extracted the lower-order components ranging from Z_0 to Z_9 and higher-order components
 ranging from Z_{10} and higher. Note that the Zernike polynomial used here follows the
 OSA/ANSI standard index. Figures R3a and d show the original aberration maps obtained from
 the data taken at a depth of 120 μm beneath the dura. Figures R3b and e are their respective
 lower-order components, and Figs. R3c and f are their respective higher-order components. We
 then compared the reconstructed images by correcting the original aberration maps (Fig. R3g),
 only the lower-order components (Fig. R3h), and only the higher-order components (Fig. R3i).

The fact that the fine myelin segment appears only with the correction of the higher-order
 components supports that the fine phase patterns are the major factors responsible for image
 reconstruction. This is true for the data taken at a depth of 550 μm below the dura as well
 (Fig. R4).

**Figure R3. Contribution of low-order and high-order aberrations to image reconstruction at layer**
 **1.** The data acquired at layer 1 at a depth of 120 μm beneath the dura was analyzed. **a-c**, Input aberration
 maps applied to reconstruct the images (**a**) with the aberration found with the conjugate-CLASS
 algorithm, (**b**) with only low order aberration decomposed by Zernike polynomial, and (**c**) with the high
 order aberration. Scale bar, 80 μm . **d-f**, Same as **a-c**, but for the output aberration maps. **g-i**,
 Reconstructed MIP images through the depth from 104 μm to 136 μm . Scale bar, 20 μm .

**Figure R4. Contribution of low-order and high-order aberrations to image reconstruction at layer**
 **4.** The data acquired at layer 4 at a depth of 550 μm beneath the dura was analyzed. **a-c,** Input
 aberration maps applied to reconstruct the images (**a**) with the aberration found with the conjugate-
 CLASS algorithm, (**b**) with only low order aberration decomposed by Zernike polynomial, and (**c**) with
 the high order aberration. Scale bar, 200 μm . **d-f,** Same as **a-c,** but for the output aberration maps. **g-i,**
 Reconstructed MIP images through the depth from 534 μm to 566 μm . Scale bar, 20 μm .

At this point, we would like to clarify the difference in the physical meaning of the aberration
 map between pupil-CLASS and conjugate-CLASS (Fig. R5a). The aberration map obtained by
 the pupil-CLASS is the angle-dependent phase retardation. On the contrary, that obtained by
 the conjugate-CLASS is the local phase variations induced by the skull at the conjugate plane.
 Therefore, the fine phase patterns in the aberration map are associated with the fine spatial
 structures of the skull. The slowly varying phase pattern can be due to the slowly varying
 curvature of the skull. It is not legitimate to conduct the Zernike decomposition and find the
 defocus component directly from the conjugate-CLASS aberration map.

To find the defocus component, it is necessary to convert the aberration map obtained by the
 conjugate-CLASS into that in the pupil plane. This is done by the following procedures. The
 aberration map in the conjugate plane is applied to a focused illumination at the conjugate plane.
 This modified wave is propagated to the object plane, and its inverse Fourier transform is taken
 to obtain the aberration map in the pupil plane. Figure R5c shows thus obtained pupil aberration
 map from the conjugate aberration map in Figure R5b. We then perform the Zernike
 decomposition of the pupil aberration map (Figs. R5d and e) and found that the defocus
 aberration accounts for less than 2 % of the total wavefront distortion. As such, the defocus
 correction made little effect on the Strehl ratio and, thus, the expected enhancement estimated
 by the inverse of the Strehl ratio (Fig. R6).

**Figure R5. Conversion of the aberration map at the skull plane to that in the pupil plane.**
 Schematic of the wave propagation to the mouse brain through the skull. The conjugate-CLASS
 algorithm corrects the aberration for the dark blue region at the skull plane marked in the schematic.
 Therefore, the aberration map corresponds to the effective spatial phase retardation induced by the skull.
 On the contrary, pupil aberration marked with dark blue at the pupil plane in the schematic accounts for
 the angle-dependent phase retardation. **b**, conjugate-CLASS aberration maps acquired for three
 representative data. **c**, Pupil aberration maps obtained by converting the conjugate aberration maps in
 **b**. **d**, Pupil aberration maps after removing the defocus component, which is the fourth index of Zernike
 polynomial Z_4 , in **c**. **e**, Zernike coefficients of the pupil aberration in **c**. Z_4 component accounts for
 0.55 %, 1.9 %, and 0.09 % of the total wavefront distortion from the top.

**Figure R6. Strehl ratio depending on the depth before and after Z_4 component subtraction.** **a,**
 Strehl ratio before (red dots) and after (blue dots) Z_4 component subtraction. The data is from
 Supplementary Movie 3. **b,** Same as **a**, but for the data in the main text Fig. 2. For data #2, we only took
 data in the cortical layers 1 and 4 to focus on the myelin analysis on those layers. **c,** Expected
 enhancement of the PSF intensity by the aberration correction for the mouse #1 data before (red dots)
 and after (blue dots). Enhancement was estimated by the inverse of the Strehl ratio. **d,** Same as **c**, but
 for data #2.

We replaced the original Strehl ratio analysis with these new analyses. We also added the
 following sentences to the discussion section. The newly analyzed data were added to
 Supplementary section 4.3.

“It should be noted that the physical meaning of the aberration map in conjugate-CLASS differs
 from that of the pupil-CLASS. The aberration map of the conjugate-CLASS corresponds to the
 effective spatial phase retardation in the plane conjugate to the skull, while that of the pupil-
 CLASS is the angle-dependent phase retardation. Their direct comparison requires the
 conversion of the aberration map from one basis to the other (Supplementary section 4.3).”

*7/ This defocus that grows with z also explains why “We observed that the signal*
 *enhancement after the aberration correction grew exponentially with depth”. This signal*
 *enhancement is therefore somehow artificial since it could have been a priori compensated*

*in their algorithm. The same remark holds for the following sentence: “The PSF width was*
*broadened by 44 times relative to the diffraction limit at cortical layer 4, supporting the*
*difficulty in reaching cortical layer 4 via the intact skull.”. I think there might be an*
*overstatement here. State-of-the-art 3D imaging techniques, such as confocal microscopy or*
*OCT, can compensate for such a defocus. Hence I think the impact of defocus and skull-*
*aberrations should be discriminated.*

Our detailed analysis in response to the reviewer’s comment #6 addressed this reviewer’s
remark.

*8/ Related to this question of defocus, would that be a better option to measure the reflection*
*matrix directly in the (win,wout) basis? In that case, the authors could have used the CLASS*
*algorithm directly on the recorded reflection matrix. I think it would be nice to discuss this*
*option and argue about its advantages/defaults compared to the authors’ choice to measure*
*the reflection matrix in a plane conjugated with the focal plane of the microscope objective.*

This is an interesting point. We currently acquire the reflection matrix by setting the objective
focus to the cortical layers underneath the skull with the reference beam path matched to the
focal plane. As the reviewer pointed out, we can fix the objective focus to the skull plane and
scan only the time-gating window throughout the cortical layers for the volumetric image
acquisition. This procedure can have potential benefits in that there is no need to scan the
objective focus and convert the basis of the matrix. However, the signal collection efficiency
will be substantially reduced with the increase of the target depth due to the defocus from the
objective focal plane. Furthermore, strong reflection from the skull will occupy most of the
detector dynamic range. For these reasons, we think that the current collection geometry is
ideal for detecting weak reflection signals deep from the brain tissues.

We added the following sentences to the discussion section.

“The present study acquired the reflection matrix by setting the objective focus to the cortical
layers underneath the skull with the reference beam path matched to the focal plane. One can
also fix the objective focus to the skull plane and scan only the time-gating window throughout
the cortical layers to obtain reflection matrices for the volumetric image acquisition. This
procedure can have potential benefits in that there is no need to scan the objective focus and
convert the basis of the matrix. However, the signal collection efficiency can be substantially
reduced with the increase of the target depth due to the defocus from the objective focal plane.”

*9/ Do the aberration maps displayed in Fig.2d correspond to input or output aberration phase*
*laws? Could the authors show a comparison between input and output aberration phase laws*
*to see whether they differ or not. Normally they should not because of spatial reciprocity but*
*I guess this property is also extremely sensitive to alignment issues and other imperfections*
*of the imaging system.*

As the reviewer pointed out, the input and output aberrations maps should be identical in theory
due to reciprocity. For this reason, we showed only the output aberration map in Fig. 2d. Here
in Fig. R7, we show both the input and output aberration maps. They are almost identical with
a correlation of 0.75 or higher. As the reviewer pointed out, there was a minor difference due
to alignment issues and other imperfections of the imaging system.

We added Fig. R7 to Supplementary section 1.5 and inserted the following sentence into the
caption of Fig. 2d.

“Only the output aberration maps are shown here, and the associated input aberrations are
shown in Supplementary Fig. 4.”

**Figure R7. Typical input and output aberration maps. a**, Aberration maps at a depth of 60 μm
beneath the dura. The correlation between the input and output aberration maps was 0.75. **b**, Same as **a**,
but at a depth of 150 μm beneath the dura. Correlation value: 0.79. **c**, Same as **a**, but at a depth of 450
360 μm beneath the dura. Correlation value: 0.80. Scale bars, 80 μm .

*10/ When discussing the merit of their CC-AO algorithm with respect to CLASS, the authors*
*mention that in the original CLASS “There is also degeneracy in the tilt and defocus in the*
*pupil AO, which can give rise to the lateral and axial shifts of reconstructed images in the*
*individual patches”. I’m not sure to understand what they mean by “degeneracy”. Can the*
*authors be more explicit?*

The original pupil-CLASS algorithm finds the aberration map that maximizes the total single
scattering intensity of the reconstructed image. In this process, there can be multiple solutions
that can provide the same total intensity. Suppose the aberration map has an additional phase-
tilt component with respect to the ground truth map. Then, the object spectrum can have the
opposite tilt component to counterbalance the tilt in the aberration map in such a way that the
resulting reflection matrix is the same as the original reflection matrix. In other words, an
aberration map with an overall phase tilt causing a lateral image shift could be another solution
to the algorithm.

The tilt degeneracy can cause difficulty in merging reconstructed image patches. Due to the
small patch size in the pupil-CLASS, many small image patches as large as 16×16 should be
merged to form an image over the full ROI. However, the unpredictable lateral image shift in
each patch makes the merged image fragmented such that the reconstructed myelin fiber can
be disconnected after merging. One can manually find and correct the tilt component in each
aberration map. This is a substantial work by itself, and its precision cannot be guaranteed when
the aberration map is too complex. The tilt degeneracy can also occur in the conjugate-CLASS
algorithm, but this is not a big issue since the patch size is large enough to faithfully connect
multiple patches.

In the case of defocus, it is an ambiguity rather than a degeneracy. When the myelin segments
are located off from the objective focus, the algorithm tends to add a defocus in the aberration
map to maximize the reconstructed image intensity. In this case, the focus of each patch can be
shifted to the plane where the myelin is situated. This can also cause the fragmentation of the
merged image in the pupil-CLASS especially when the myelin segment is slanted along the

axial direction. The ambiguity in the defocus can also occur in the conjugate-CLASS, but the
large isoplanatic patch size ensures the fidelity of merging subregion images.

We added these clarifications to Supplementary section 4.4.

*I congratulate the authors for their work and hope they will find my comments useful and*
*relevant.*

Once again, we deeply appreciate the reviewer's valuable comments. We found them truly
helpful in improving the integrity of our work.

**Reviewer #2**

*The authors present a novel development of their optical coherence microscopy-related*
*adaptive optics method designed to amplify the signal from the back-reflected single-*
*scattered waves inside biological media. Originally described as CASS (Kang et al., Nat Phot*
*2015), it allows one to increase the signal from waves back-scattered from the target inside*
*a scattering medium, while decreasing the contribution of the multiple-scattered light at the*
*same time through the use of time gating and coherent summation of the back-reflected*
*waves. Further developments (termed CLASS, Kang et al., Nat Comm 2017) allowed the*
*authors to correct aberrations separately for the light scattered on the way to and away from*
*the imaging plane. Having already demonstrated its utility in imaging myelin fibers through*
*intact skulls in a mouse (Yoon et al., Nat Comm 2020), in the present study the authors*
*expand this technique through the use of computational conjugate adaptive optics to*
*increase the field of view after aberration correction. As the majority of aberrations comes*
*from the intact skull, correcting the aberrations in the skull plane allowed authors to image*
*myelin fibers deeper inside the mouse cortex with larger FOV and better signal-to-noise ratio.*
*Furthermore, the computational propagation to the conjugate plane helped avoid time-*
*consuming alignment of the setup for hardware conjugate AO. The authors also changed*
*the wavelength from 900 nm to 1300 nm to reduce scattering.*

*I found this study to be an interesting development of the imaging methods based on single-*
*scattered waves. However, I would like the authors to address the following issues before it*
*can be published in Nature Communications:*

We appreciate the reviewer's clear summary of the progress made in our group over the past
several years and acknowledging the value of the newly proposed method in the present study.
In this revision, we carefully addressed the reviewer's valuable comments and suggestions.

*– (1) The data processing is insufficiently described in the paper, even if it was described in*
*the previous studies. Why did the authors change notation vs. their previous papers? Data*
*analysis steps with a corresponding data analysis pipeline should be demonstrated in the*
*Supplementary methods.*

The data processing procedures were rather brief in our original manuscript. Following the
reviewer's suggestion, we added detailed procedures to Supplementary section 3.

As the reviewer is well aware, the newly proposed conjugate-CLASS algorithm finds the
aberration maps in the plane $(\mathbf{w}_0, \mathbf{w}_h)$ conjugate to the mouse skull while the previous pupil-
CLASS finds aberration maps in the pupil plane $(\mathbf{k}_o, \mathbf{k}_h)$ associated with the object plane
$(\mathbf{r}_o, \mathbf{r}_h)$. Note that we renamed the term CC-AO as conjugate-CLASS following the reviewer's
suggestion. Since the reflection matrix is recorded on the $(\mathbf{r}_o, \mathbf{r}_h)$ basis in the experiment, the
basis of the reflection matrix should be converted to $(\mathbf{w}_0, \mathbf{w}_h)$ basis, and the theoretical
description of the conjugate-CLASS algorithm should be reformulated in this new basis. Please
find the details of the basis conversion and the theoretical description of the algorithm in
Supplementary section 2.

*– (2) What are the conditions for convergence of their method? How do authors show that*
*maximum intensity in the imaging place results in a correct estimation of the aberration*

*field? Is sparsity of the sample and strong back-scattering of the target structures an absolute*
 *prerequisite for this method to converge? Would this limit the applicability for other types of*
 *samples?*

The convergence condition for the pupil-CLASS is well documented in our earlier studies.
 Here we applied a similar description to the conjugate-CLASS framework. In Supplementary
 section 2, we describe the electric field propagated to the plane conjugate to the skull in the
 following equation,

$$447 \quad E(\mathbf{W}; \mathbf{w}_n) = -\frac{e^{2kz}}{\lambda^2 z^2} e^{i\phi_o(\mathbf{W}-\mathbf{w}_n)} \tilde{O}_M(k\mathbf{W}/z) e^{i\phi_n(\mathbf{w}_n)} + E_M(\mathbf{W}; \mathbf{w}_n),$$

where $\mathbf{W} = \mathbf{w}_o + \mathbf{w}_n$. The correlation of output waves between the two representative
 incident wave vectors $\mathbf{w}_n^{(l)}$ and $\mathbf{w}_n^{(m)}$ can be expressed as $\langle E(\mathbf{W}; \mathbf{w}_n^{(l)}) E^*(\mathbf{W}; \mathbf{w}_n^{(m)}) \rangle_{\mathbf{W}}$.
 This can be expanded as follows.

$$451 \quad \langle E(\mathbf{W}; \mathbf{w}_n^{(l)}) E^*(\mathbf{W}; \mathbf{w}_n^{(m)}) \rangle_{\mathbf{W}}$$

$$452 \quad \approx e^{i[\phi_n(\mathbf{w}_n^{(l)}) - \phi_n(\mathbf{w}_n^{(m)})]} \alpha(z) \langle |\tilde{O}_M(k\mathbf{W}/z)|^2 e^{i[\phi_o(\mathbf{W}-\mathbf{w}_n^{(l)}) - \phi_o(\mathbf{W}-\mathbf{w}_n^{(m)})]} \rangle_{\mathbf{W}}$$

$$453 \quad + \langle E_M(\mathbf{W}; \mathbf{w}_n^{(l)}) E_M^*(\mathbf{W}; \mathbf{w}_n^{(m)}) \rangle_{\mathbf{W}},$$

where $\alpha(z) = \left(\frac{e^{2kz}}{\lambda^2 z^2}\right)^2$. We ignored the cross terms since they are much smaller than the
 second term on the right-hand side. The factor $e^{i[\phi_n(\mathbf{w}_n^{(l)}) - \phi_n(\mathbf{w}_n^{(m)})]}$ in the first-term on the
 right-hand side is the input aberration that we are going to identify in this correlation, and the
 second term serves as noise. The magnitude of first term is largely given by single-scattering
 intensity $I_S \approx \alpha(z) |\tilde{O}_M(k\mathbf{W}/z)|^2$ and the normalized cross-correlation of output aberrations
 $\xi \approx \left| \langle e^{i[\phi_o(\mathbf{W}-\mathbf{w}_n^{(l)}) - \phi_o(\mathbf{W}-\mathbf{w}_n^{(m)})]} \rangle_{\mathbf{W}} \right| / N(\mathbf{W})$, where $N(\mathbf{W})$ is the number of channel used to
 average over \mathbf{W} . Since the magnitude of ξ is related to the degree of aberration, it gets smaller
 as the aberration becomes complex. The magnitude of the second term is given approximately
 by $I_M \sqrt{N(\mathbf{W})}$. Hence, the fidelity of correction is determined by the ratio between the first and
 the second terms, $\chi = \xi (I_S / I_M) \sqrt{N(\mathbf{W})}$. The algorithm's convergence depends on the certain
 threshold value of χ that is given by single to multiple scattering intensity ratio I_S / I_M , the
 degree of aberration ξ , and the number of channels $N(\mathbf{W})$.

The discussion above can explain the next question. Since our algorithm's correction fidelity
 depends on a single to multiple scattering intensity ratio, it requires a certain level of back-
 scattering from the target sample. However, there is no requirement about the sparsity of the
 sample as the fidelity is determined only by the total intensity.

We added this discussion on the convergence condition to Supplementary section 2.3.4.

– (3) *Prior related work in ultrasound not referenced or even mentioned.*

Our study focuses on deep-brain myelin imaging requiring high spatial resolution (~1 micron).
Thus, we confined the reference coverage mainly to optical imaging. However, we think that
it is worthwhile to introduce ultrasound imaging works to the discussion section because the
newly developed algorithm can be applied to the reflective matrices acquired in ultrasound
imaging as well. As such, we added the following relevant papers and sentence to the discussion
section.

Aubry, A. & Derode, A. Random matrix theory applied to acoustic backscattering and
imaging in complex media. *Phys. Rev. Lett.* **102**, 084301 (2008).

Lambert, W., Cobus, L.A., Frappart, T., Fink, M. & Aubry, A. Distortion matrix approach for
ultrasound imaging of random scattering media. *Proc. Natl. Acad. Sci.* **117**, 14645-14656
(2020).

“Our work can also be extended to ultrasound imaging where the reflection matrix is
routinely measured^{24, 25}.”

– (4) *The authors compare their method to optical coherence microscopy in the time domain.*
*However, in many applications OCM/OCT is used in frequency domain due to the increased*
*SNR. How would their method compare to the frequency-domain OCM? What would be the*
*benefits in terms of SNR and imaging depth? How much faster/slower would the imaging*
*be?*

We think that the direct comparison of our method with either time-domain or frequency-
domain OCM is relevant only when they are equipped with adaptive optics capability. The
benefit of our method with respect to various adaptive optics OCM methods is well
documented in our earlier papers. Our method is an extension of the time-domain OCM. We
record the electric field map of the backscattered wave for each focus illumination while the
conventional time-domain OCM records the electric field only at a position conjugate to the
illumination focus. While the acquisition of the full information makes the imaging speed
slower, it enables us to computationally correct both the input and output aberration maps.

One can consider the extension of the frequency-domain OCT/OCM to our reflection matrix
approach. In fact, the spectrometer-based FD-OCT is not applicable to our reflection matrix
approach. The 2D spatial mapping of the backscattered waves is not possible because one
spatial degree of the camera should be assigned to the spectral recording. On the contrary, the
swept-source-based FD-OCT can be extended to our reflection matrix approach. In this case,
the 2D camera can record all the backscattered waves while scanning both the illumination
position and wavelength. While this is possible in principle, it will take a huge amount of time
to record a single set of data. However, once recorded, it will open a new opportunity to correct
both the spatial aberrations and spectral dispersions.

In summary, the conventional TD- and FD-OCM are suitable for applications where real-time
imaging is required and the degree of aberrations is not substantial. On the contrary, our
approach can be employed for severely scattering and aberrating conditions such as through-
skull imaging where conventional OCM fails to work.

– (5) *Inconsistent method nomenclature. Why not pupil CLASS vs. conjugate CLASS as*

*method names? Authors use these, RMM and CC-AO interchangeably, which is confusing,*
*even if they reference different parts of their methods.*

We acknowledge the reviewer's concern about the nomenclature and appreciate a valuable
suggestion. The term reflection matrix microscopy (RMM) describes the experimental part of
our method, and conjugate-CLASS describes the newly proposed algorithm. Since RMM also
applies to earlier works exploiting the reflection matrix, either CC-AO or conjugate CLASS
represents the present work better. We think that the reviewer's suggestion of naming the
present work as conjugate-CLASS and earlier work as pupil-CLASS is an excellent idea. It
makes a good contrast with the earlier studies while clearly capturing the essence of the present
study. As such, we used the term conjugate-CLASS throughout this revision.

– (6) *From the text and mathematical derivations it follows that the skull plane doesn't need*
*to be physically conjugate but in Fig. 1c it is. Fig. should reflect it clearly.*

As the reviewer mentioned, Fig.1c is a hypothetical experimental schematic describing the
basis of the conjugate-plane reflection matrix R_{con} , not the real experimental setup. Following
the reviewer's suggestion, we added the following sentence to the caption of Fig. 1c.

“Note that this schematic is a hypothetical experimental setup describing the basis of R_{con} ,
not the real experimental setup.”

– (7) *Not all components in Fig 1a are labelled (pinhole, beam splitter in front of PD). A list*
*of lenses/components used would be useful for reproducibility.*

We appreciate the reviewer's suggestion. We labeled all components in Fig. 1a and described
their full names in the figure caption. A list of full components is now available in
Supplementary section 1.

– (8) *Please check the language. For example: use of “ingenious” seems misplaced.*

We revised the text as follows.

“On the contrary, our conjugate-CLASS can freely choose the optimal conjugate plane after
the data acquisition and administrate a large number of iterations computationally to cope with
much more complex aberrations.”

We also checked the language of the manuscript again for better readability.

**Reviewer #3**

*This group has significantly improved their original CLASS system and algorithm which*
*originally performed computational adaptive optics at 900 nm. First, they built a system at*
*1300 nm to take advantage of reduced scattering in the brain and skull, and second, they*
*applied their algorithm in the plane of the skull rather than the pupil plane of the sample.*
*As the major aberrations occur in the skull, according to their claim, this gives them a larger*
*isoplanatic patch and simplifies image reconstruction. Myelinated axons were imaged in*
*layer 4 in young mice over time with apparently high resolution.*

We appreciate the reviewer for carefully reading our manuscript and providing us with valuable
comments. In the following, we have addressed all reviewer's concerns and questions.

*I have the following criticisms which should be addressed in the manuscript text:*

*1) Younger mice have thinner skulls which are less scattering and also less myelin making*
*it easier to see deeper. The mice imaged here are not yet adults and are still developing*
*myelination. How the system will perform in adult or aged (years old) mice is not clear.*

We agree with the reviewer in that our system could conduct through-skull imaging of mice in
the developing stages. According to the literature, mice older than 12 weeks are usually
considered in their adulthood. Although there are variations depending on individual mice, our
system could image cortex layer 1 for mice up to 10 weeks old, close to adulthood. However,
we could image mice up to 7 weeks old in the case of cortex layer 4. On the one hand, we
expect a further increase in the source wavelength to 1700 nm [Ref. 7. Zhu, J. *et al.*, Light Sci.
Appl. 10, 145 (2021)] will help to reach cortical layer 4 of the adult mice. On the other hand,
the performance of the present system is good enough to take the main benefit of through-skull
imaging, which is to observe the myelination process in young individuals that are highly
sensitive to cranial surgery.

Following the reviewer's suggestion, we specified in the revised main text that the current
system investigated mice in the developing stages close to adulthood. We added the following
description to the discussion section.

"Our current system could conduct through-skull imaging of mice in the developing stages.
Although there are variations depending on individual mice, our system could image cortex
layer 1 for up to 10 weeks, close to adulthood. However, we could image mice about 7 weeks
old in the case of cortex layer 4. We expect that a further increase in the source wavelength to
1700 nm will help to reach cortical layer 4 of the adult mice⁷."

*2) The SHG inset in Fig. 2A has a depth axis that appears inconsistent with the rest of the*
*figure.*

We appreciate the reviewer's comment and have corrected the figure as suggested.

*3) They talk about tracing myelin ('In summary, our 1.3- μ m RMM system and image*
*reconstruction algorithm enabled the tracing of individual myelin segments in the cortical*
*brain'), but I feel they are just monitoring individual myelinated segments, not really tracing*

*them.*

Following the reviewer's comment, we replaced "tracing" with "monitoring" in the revised
manuscript.

"In summary, our 1.3- μm RMM system and image reconstruction algorithm enabled the
monitoring of individual myelin segments in the cortical brain, without cranial surgery."

**4) Recent 1700 nm OCM work should be cited; Zhu, J., Freitas, H.R., Maezawa, I. et al.**
**1700 nm optical coherence microscopy enables minimally invasive, label-free, in vivo optical**
**biopsy deep in the mouse brain. *Light Sci Appl* 10, 145**
**(2021). <https://doi.org/10.1038/s41377-021-00586-7>**

We appreciate the suggestion of this important reference. We cited this article as reference #7
in the introduction section of the revised manuscript.

"Label-free imaging modalities, such as optical coherence microscopy (OCM)^{6,7} combining
the time gating and confocal gating, spectral confocal reflectance microscopy⁸, third-harmonic
generation microscopy⁹, and coherent Raman imaging¹⁰ have drawn special attention as they
are free from these issues."

**5) The authors state: "genetic labeling can visualize a small fraction of mature myelination**
**owing to its partial expression in a small population of oligodendrocytes"- I believe that this**
**partial expression might be by design, not by necessity, so that individual myelin axons can**
**be traced and seen.**

We agree with the reviewer in that the partial expression of the genetic labeling is designed,
not a limitation. This has been necessary to minimize the background fluorescence in deep-
tissue imaging. The full expression of oligodendrocytes can induce strong background
fluorescence, thereby impeding the observation of myelinated axons at a depth of interest. We
revised the sentence as follows in the revised manuscript.

"Sparse labeling is usually implemented in the genetic labeling to minimize background
fluorescence in deep-tissue imaging, which leads to the visualization of only a small fraction
of mature myelination."

**6) They cite Ref. 10 for the inflammatory response of skull thinning and opening but not**
**earlier work which has shown skull thinning with glass reinforcement to be significantly less**
**inflammatory than skull opening: Drew, P., Shih, A., Driscoll, J. et al. *Chronic optical access***
**through a polished and reinforced thinned skull. *Nat Methods* 7, 981–984**
**(2010). <https://doi.org/10.1038/nmeth.1530>. Moreover, a more accurate statement of Ref.**
**10 is that surgical skill is an important determinant of outcome with skull thinning, with**
**inadvertent mechanical trauma causing CNS disruptions (even this is still a good motivation**
**for the approach presented here though)**

We appreciate the reviewer's thoughtful comments and suggestions. We revised the text
accordingly as suggested.

“However, all these methods require either the removal of the skull for the implantation of a
 cranial window or thinning of the skull to secure clear optical access. These procedures can
 cause inadvertent mechanical trauma¹¹. Although the thinned skull with glass reinforcement
 can significantly minimize inflammation¹², the outcome of skull opening and thinning depends
 highly on the surgical skill.”

**7) Are the conjugate plane aberration maps in Fig. 2D covering/obscuring some of the sub-**
 **region aberration maps? Please shift to the side. Also does ‘conjugate plane’ mentioned in**
 **the caption refer to the conjugate plane of the skull? Please explicitly state. Why is the scale**
 **bar different than the others in the top map in panel d?**

In the original manuscript, the representative aberration maps were selected to provide their
 enlarged views for visibility. Since this can cause confusion, we removed them in the revised
 manuscript. The conjugate plane refers to the plane conjugate to the skull. Therefore, we
 revised the text in the caption as follows.

“The aberration maps represent the spatial phase retardation at the plane conjugate to the skull.”

The size of the scale bar in Fig. 2d varies with depth. This is because the size of the aberration
 map identified by the conjugate-CLASS (upon request of reviewer #2, we changed the term
 CC-AO to conjugate-CLASS) increases with depth due to the illumination/detection geometry.
 As shown in Fig. R8, the area in the skull plane responsible for the focused illumination is
 enlarged with the increase of the distance between the objective focus and the conjugate plane.
 The numerical aperture of the objective lens is another factor determining the aberration map
 size. In our experimental configuration, the diameter of the aberration map was given
 approximately by $D \sim 1.2z$.

**Figure R8. The size of aberration maps at the skull plane. a,** Schematic of two focused beams
 illuminating on the two ends of ROI at the sample plane through the skull. L_0 : the size of ROI in one
 axis, which is typically 160 μm in the experiment. θ_{NA} : the angle of the cone of the illumination beam.
 This can be calculated by $NA = n \times \sin\theta_{NA}$, where NA is the numerical aperture and n is the average
 refractive index of the medium. z : the distance from the sample plane to the skull plane. R : radius of
 the focused illumination in the skull plane, which can be calculated by $R = z \times \tan\theta_{NA}$. As a result,

the diameter of aberration map D can be obtained by $D = L_0 + 2R$. **b**, The diameter of aberration map
D depending on the imaging depth z . Relation between the diameter and the imaging depth was given
by $D \sim 1.2z$.

We added the following sentence to the figure caption and this discussion to Supplementary
section 1.6.

“Note that the diameter of the aberration map increases with depth due to illumination/detection
geometry (Supplementary section 1.6).”

*Also why are these aberration maps ‘representative’-I thought with the conjugate plane*
*algorithm there should be only one aberration map in the skull plane.*

The conjugate-CLASS also has a finite isoplanatic size although it is much larger than that of
the pupil-CLASS. This is because the description of a thick skull by a thin phase plate at a
conjugate plane is still an approximation. For the same experimental data, we found that the
isoplanatic size of the pupil-CLASS was around $15 \times 15 \mu\text{m}^2$ while that of our conjugate-
CLASS was around $80 \times 80 \mu\text{m}^2$ (Supplementary Fig. 8). Therefore, the size of each
subregion to be analyzed can be as large as $80 \times 80 \mu\text{m}^2$. Since the recorded ROI was
$160 \times 160 \mu\text{m}^2$, we can divide the entire ROI into 4×4 subregions with a 50 % overlapping
ratio for optimal visibility. In this case, the computation time for reconstructing the entire ROI
was 2,386 s.

In the actual data analysis, we chose the subregion size of $64 \times 64 \mu\text{m}^2$ and divided the entire
ROI into 5×5 subregions, which resulted in a reduced computation time of 1557 s. In fact,
as the patch size decreases and, thus, the number of patches increases, the total computation
time gets shorter because the computation time for each patch scales with a power of 2 with
respect to the area of the patch as shown in Fig. R9. Since we processed the data over 140 to
170 depths in the volumetric image, this difference in computation time is a critical factor. We
didn’t reduce the subregion size to smaller than $64 \times 64 \mu\text{m}^2$ to avoid image fragmentation
in merging multiple images. Note that the computation time is spent on the post-processing
and, thus, does not affect to the in-vivo imaging.

We added this discussion to Supplementary section 1.4.

**Figure R9. Computation time depending on the analysis patch area.** Data points in the figure
correspond to $40 \times 40 \mu\text{m}^2$, $46 \times 46 \mu\text{m}^2$, $52 \times 52 \mu\text{m}^2$, $64 \times 64 \mu\text{m}^2$, $80 \times 80 \mu\text{m}^2$, and
$105 \times 105 \mu\text{m}^2$. A computer used for this analysis is equipped with i9-12900KS CPU and NVIDIA
RTX A6000 GPU. Note that the computation time for the full field of view is given by the multiplication
of the computation time shown here to the number of patches. The data was well-fitted with the second-
degree polynomial function.

**8) Please explain why the reflection matrix depends so strongly on the illuminated position**
**in Fig. 1d? This suggests to me that system aberrations may be dominant, as I don't see why**
**skull aberration should be radially symmetric with respect to the field of view.**

As mentioned in the figure caption, the images in Fig. 1d are a set of recorded electric field
images $E(\mathbf{r}_o - \mathbf{r}_h ; \mathbf{r}_h)$ of the reflected waves in the camera plane for various \mathbf{r}_h , not the
reflection matrices. Therefore, they were highly dependent on the illumination position. These
images are converted to columns to form a single reflection matrix. We added a detailed
description of the data processing to Supplementary section 3.

**9) Was any effort made to measure or correct system (not brain) aberrations independently**
**of CLASS algorithm?**

We didn't measure/correct system aberration independently of the CLASS algorithm. Instead,
we verified that the CLASS algorithm finds the ground-truth aberration. To this end, we
displayed a designed phase pattern (Fig. R10a) on the SLM in the pupil plane of the sample
beam path and obtained the aberration map that it induced (Fig. R10b). From the resemblance
between the two maps, we could confirm that the CLASS algorithm finds the real aberration.
There was a slowly varying aberration pattern on the background in Fig. R10b. This turned out
to be the system aberration (Fig. R10c).

**Figure R10. Verification of CLASS algorithm performance.** **a**, Designed phase pattern displayed on
the SLM at the pupil plane. **b**, Aberration map found by the CLASS algorithm. **c**, System aberration
acquired by the CLASS algorithm when a blank pattern was displayed on the SLM.

**10) Please make clear that Fig. 1c is not the actual experimental scheme, and reflects a**
**hypothetical setup.**

We added the following sentence to the caption to clarify that Fig. 1c is 'the hypothetical setup.'

“Note that this schematic is a hypothetical experimental setup describing the basis of R_{con} ,
not the real experimental setup.”

**11) Also shouldn't both panels d and e in Fig. 1 have an amplitude and a phase? If this is**
**the case, please explain that the phase is not being shown.**

Indeed, each image has both amplitude and phase parts. We specified in the figure caption that
the phase part is not shown.

**12) Pg 2: “observed the emergence of new myelin segments in cortical layer 4 with aging.”-**
**> I would say ‘with development’ instead of ‘with aging’**

We agree with the reviewer. In the revised manuscript, we replaced “with aging” with “with
development.”

**13) They state that they are “exploiting the shift-invariance of \tilde{OM} with respect to w_{in}**
**a. Is it also shift invariant with respect to w_{out} ?**

Yes. Equation (2) tells that \tilde{O}_M is shift-invariant with respect to w_{in} from the viewpoint of
w_{O} . Likewise, \tilde{O}_M is shift-invariant with respect to w_{O} from the viewpoint of w_{in} .

**b. Does shift-invariant mean that it does not depend on w_{in} ? If true please rephrase in a**
**simpler way.**

The shift-invariance means that the functional form is maintained with the change in w_{in} . We
added the following sentence to the Methods section to clarify this point.

“In other words, the functional form of the object spectrum is maintained with the change in
w_{in} .”

**c. Can you restate this as a condition on OM (amplitude reflectance of object)?**

In fact, the shift-invariance is not a condition but an intrinsic property of the object at the focal
plane.

**14) They state ‘The aberration-corrected images were acquired at a depth interval of 3 μm**
**from the cortical surface to a depth of $\sim 100 \mu\text{m}$ of all the mice’-was imaging not performed**
**to layer IV in some mice?**

We acknowledge that the original sentence was misleading. We performed imaging up to layer
4 for all the mice investigated in our study, but the images from the cortical surface to a depth

of ~100 μm were used for analyzing the myelin segment length in Figs. 3d and e. We revised
the sentence as follows.

“To measure the length of myelin segments in cortical layer 1 (Figs. 3d and e), we analyzed the
aberration-corrected images from the cortical surface to a depth of ~100 μm of all the mice
investigated in this study.”

***15) Their algorithm remains conceptually similar to CLASS (as I understand they apply a***
***numerical propagation to a different plane prior to applying their algorithm). It appears that***
***most of the improvement is coming from the longer wavelength used in this study. The***
***comparison between 900 nm and 1.3 microns merits more attention.***

Indeed, the use of the 1.3 μm wavelength source made a major contribution to the increase of
imaging depth due to the substantial reduction of multiple scattering and aberration. This is in
line with the earlier studies demonstrating the benefit of using the 1.3 μm wavelength source
[Ref. 6. Srinivasan, V. J., *et al.*, Opt. Express **20**, 2220 (2012)]. Specifically, we could barely
reconstruct the myelin fiber right beneath the skull by the previous 900 nm system with the
pupil-CLASS algorithm. On the contrary, we could image myelin segments at cortical layers
4-5 with the new 1.3 μm system even when the same pupil-CLASS algorithm was used. While
it will be ideal to directly compare imaging the same specimen with 900 nm and 1.3 μm systems,
this is not possible in the current system as the optics can support only the wavelengths longer
than 1.05 μm wavelength.

We would also like to emphasize that the newly developed algorithm made another important
contribution. While the previous pupil-CLASS algorithm can also visualize the myelin at 4-5
layers, the isoplanatic patch size was so small (~ 15 \times 15 μm^2), making it difficult to properly
merge subregion images to form a full field-of-view (FOV) image. In particular, the degeneracy
in tilt and the ambiguity in the defocus in reconstructing each patch make the MIP image of the
pupil-CLASS blurred and fragmented (Supplementary Fig. 9). On the contrary, the conjugate-
CLASS algorithm enlarges the isoplanatic patch size as large as 80 \times 80 μm^2 (Supplementary
Fig. 8), allowing us to form the full FOV image with high fidelity. This has been particularly
important for quantitatively assessing the statistics of myelin segment length.

We added the benefit of 1.3 μm source to the discussion section.

“The use of the 1.3 μm wavelength source made a major contribution to the increase of imaging
depth due to the substantial reduction of multiple scattering and aberration. This is in line with
the earlier studies demonstrating the benefit of using the 1.3 μm wavelength source⁶.
Specifically, we could barely reconstruct the myelin fiber right beneath the skull by the
previous 900 nm system with the pupil-CLASS algorithm. On the contrary, we could image
myelin segments at layers 4-5 with the new 1.3 μm system even when the same pupil-CLASS
algorithm was used.”

***16) In the Supplement Section 2.3 it is not at all clear if they have made any modifications***
***to their iterative algorithm relative to prior CLASS works-please explicitly state if there are***
***differences, or cite the prior work if not.***

The major modification of the conjugate-CLASS compared to the pupil-CLASS is the
 numerical propagation of the basis of the reflection matrix to the conjugate (skull) plane. This
 requires the reformulation of the matrix description as given in Supplementary sections 2.1 and
 2.2. Since the functional form of the matrix decomposition of \mathbf{R}_{con} resembles that of the
 original matrix \mathbf{R} , we could use a similar iterative algorithm engine as before although the
 physical interpretations of the optimized results are different. We made this point clear by
 adding the following sentences to Supplementary section 2.3.

“Here, we exploit the fact that the object spectrum $\tilde{O}_M(k(\mathbf{w}_o + \mathbf{w}_h)/z)$ in $E(\mathbf{w}_o; \mathbf{w}_h)$ is
 shift-invariant with respect to \mathbf{w}_h . Since this resembles the shift-invariance of $\tilde{O}(\mathbf{k}_o - \mathbf{k}_h)$
 with respect to \mathbf{k}_h in the pupil-CLASS, we used a similar iterative optimization engine as
 that of the pupil-CLASS.”

*17) I don't understand why phi_in and phi_out don't have some relationship and are treated*
 *as totally independent-Aren't these double pass aberrations? Are all aberration maps (e.g.*
 *2d) related to phi_in? What about phi_out?*

As the reviewer pointed out, the ϕ_h and ϕ_o should be the same due to the double-pass
 geometry and reciprocity. However, they can slightly be different due to experimental
 imperfections such as the slight misalignment between the illumination and detection pathways.
 Oftentimes, this misalignment is necessary to avoid direct back-reflection from the optics.

In Fig. 2d, we showed only the output aberration maps as the input aberration maps are similar.
 Here we showed both the input and output aberration maps for a few different depths. The
 patterns have high similarity with their correlation value of 0.75 or higher.

**Figure R11. Typical input and output aberration maps. a**, Aberration maps at a depth of 60 μm
 beneath the dura. The correlation between the input and output aberration maps was 0.75. **b**, Same as **a**,
 but at a depth of 150 μm beneath the dura. Correlation value: 0.79. **c**, Same as **a**, but at a depth of 450
 823 μm beneath the dura. Correlation value: 0.80. Scale bars, 80 μm .

We added Fig. R11 to Supplementary section 1.5 and inserted the following sentence into the
 caption of Fig. 2d.

“Only the output aberration maps are shown here, and the associated input aberrations are
 shown in Supplementary Fig. 4.”

We would like to emphasize that the independent treatment of the ϕ_h and ϕ_o is an essential
 feature of the iterative algorithm. Suppose $\phi_h(\mathbf{w}_h)$ and $\phi_o(\mathbf{w}_o)$ are the same in the master
 equation in the following.

$$E(\mathbf{w}_o; \mathbf{w}_h) = -\frac{e^{2kz}}{\lambda^2 z^2} e^{i\phi_o(\mathbf{w}_o)} \tilde{O}_M \left(\frac{k}{z} (\mathbf{w}_o + \mathbf{w}_h) \right) e^{i\phi_h(\mathbf{w}_h)}.$$

If we aim to find the input aberration $\phi_h(\mathbf{w}_h)$, we need to modify the output aberration
 $\phi_o(\mathbf{w}_o) = \phi_h(\mathbf{w}_h)$ simultaneously. In other words, changing one variable affects to both the
 input and output aberrations, making the response of the $E(\mathbf{w}_o; \mathbf{w}_h)$ complicated. In fact, this
 is the reason why the finding of one-way aberration out of the round-trip measurement is
 difficult. In our CLASS algorithm, we first find approximate ϕ_h while leaving ϕ_o
 untouched. This approach enables us to exploit the correlation among the columns in the
 reflection matrix to find the approximate ϕ_h . The repeat of the similar process for finding the
 approximate ϕ_o , and the iterations of these input and output corrections ultimately lead to the
 separate finding of ϕ_h and ϕ_o .

To make this point clear, we added the following sentences to the main text explaining Eq. (1).

“In theory, ϕ_h and ϕ_o are the same due to the reciprocity, but we treat them independently
 to find ϕ_h and ϕ_o iteratively by exploiting the wave correlation in the reflection matrix.
 This also accounts for the potential mismatch between ϕ_h and ϕ_o due to experimental
 imperfections such as a slight mismatch between the illumination and detection pathways.”

REVIEWER COMMENTS

Reviewer #1 (Remarks to the Author):

I am convinced by the response of the authors and the changes made to the manuscript. The paper can be published as it is.

A. Aubry

Reviewer #2 (Remarks to the Author):

The authors addressed all of my questions and I support publication.

Reviewer #3 (Remarks to the Author):

Figure R1: Correlation is complex as described in the expression and the phase needs to be shown or accounted for somehow.

For 'w_in'-the 'i' appears missing throughout.

Reviewer #1:

I am convinced by the response of the authors and the changes made to the manuscript. The paper can be published as it is

Once again, we appreciate the reviewer's thoughtful comments and suggestions.

Reviewer #2:

The authors addressed all of my questions and I support publication.

We thank the reviewer for the constructive suggestions, which helped us to improve the scientific integrity of the manuscript.

Reviewer #3:

1) Figure R1: Correlation is complex as described in the expression and the phase needs to be shown or accounted for somehow.

We thank the reviewer for the careful reading of our manuscript and for providing thoughtful suggestions.

Figure R1 shows the autocorrelation $g_2(\tau)$ of the speckle intensity of the measured complex field maps. Therefore, the correlation in Fig. R1 is a real number. In the revised manuscript, we clarified that the correlation is the intensity correlation, not the field correlation.

2) For 'w_in'-the 'i' appears missing throughout.

We thank the reviewer for the careful reading of our manuscript. We realized that 'i's became missing while converting from a word file to a PDF file. We corrected this issue in our revision.